

# Comparison of the Coastal and Regional Ocean Community Model (CROCO) and NCAR-LES in Non-hydrostatic Simulations

Xiaoyu Fan[a], Baylor Fox-Kemper[a], Nobuhiro Suzuki[b], Qing Li[c], Patrick Marchesiello[d], Francis Auclair[e], Peter Sullivan[f], and Paul Hall[a]

[a]Department of Earth, Environmental, and Planetary Sciences, Brown University, Providence, RI 02912, USA
[b]Institute of Coastal Ocean Dynamics, Helmholtz-Zentrum Hereon
[c]Earth, Ocean and Atmospheric Sciences Thrust, The Hong Kong University of Science and Technology (Guangzhou), Guangzhou, 511400, Guangdong, China
[d]Institute of Research for Development, LEGOS
[e]Laboratoire d'Aérologie
[f]NCAR

**Correspondence:** Xiaoyu Fan (xiaoyu_fan@alumni.brown.edu)

**Abstract.**

Advances in coastal modeling and computation provide the opportunity for examining non-hydrostatic and compressible fluid effects at very small scales, but the cost of these new capabilities and the accuracy of these models versus trusted non-hydrostatic codes has yet to be determined. Here the Coastal and Regional Ocean COmmunity model (CROCO) and the NCAR

Large-Eddy Simulations (NCAR-LES) code base are compared with a focus on their simulation accuracy and computational efficiency. These models differ significantly in numerics and capabilities, so they are run on common classic problems of surface-forced, boundary-layer turbulence. In accuracy, we compare turbulence statistics, including the effect of the explicit sub-grid scale (SGS) parameterization, the effect of the second (dilatational) viscosity and the sensitivity to the speed-of-sound, which is used as part of the CROCO compressible turbulence formulation. To gauge how far CROCO is from the NCAR-LES,

we first compare the NCAR-LES with two other LES codes (PALM and Oceanigans). To judge efficiency of CROCO, strong and weak scaling simulation sets vary different problem sizes and workload per processor, respectively. Additionally, the effects of 2D decomposition of CROCO and NCAR-LES and supercomputer settings are tested. In sum, the accuracy comparison between CROCO and the NCAR-LES is similar to the NCAR-LES versus other LES codes. However, the additional capabilities of CROCO (e.g., nesting and realism) and its compressible turbulence formulation come with roughly an order of magnitude

of additional costs despite efforts to reduce them by adjusting the second viscosity and sound speed as far as accuracy allows.

## 1   Introduction

Coastal ocean modeling using limited domain sizes and open boundaries has been a standard practice for decades (Mellor, 1998; Haidvogel et al., 2008). As computing power has increased, the opportunity to simulate conditions that exceed the limits for standard oceanographic model approximations (Fox-Kemper et al., 2019) have arisen in the coastal modeling context (e.g.,

Boussinesq incompressibility, hydrostasy, the traditional approximation of the Coriolis force). Sharp topographic features,





strong internal waves and submesoscales, boundary layer turbulence, sea level rise and ice-ocean phase transitions, and many other phenomena of coastal interest could be more directly simulated with these assumptions relaxed, rather than relying on parameterizations or numerical fixes to approximate the impacts of smaller scales. By contrast, Large Eddy Simulation codes have long been used that do not make the hydrostatic approximation to study three-dimensional turbulence, but these codes often rely on numerical approaches (e.g., Fourier spectral methods) that make them unable to handle realistic topography and other aspects of coastal modeling. This paper is an evaluation of these different types of codes side-by-side on problems where they can be directly compared for accuracy and efficiency.

The Coastal and Regional Ocean Community model (CROCO) is a modelling platform for the regional and coastal ocean primarily supported by French National Research Institute for Sustainable Development (IRD) and the National Institute for Research in Digital Science and Technology (INRIA). Built on ROMS_AGRIF and the non-hydrostatic kernel of SNH, CROCO has the objective to resolve problems of very fine-scale coastal areas through nesting while at the same time operating as a standard coarse resolution coastal modeling system (Debreu et al., 2016). Activating the non-Boussinesq and non-hydrostatic kernel (NBQ) of CROCO is the precondition to solve the compressible and non-hydrostatic Navier-Stokes equations, allowing direct simulation of complex non-hydrostatic physical problems such as overturning and three-dimensional turbulence. Non-hydrostatic effects become important when the horizontal and vertical scales of motion are similar (Wedi and Smolarkiewicz, 2009; Fox-Kemper et al., 2019), and they are required in the study of small-scale phenomena in the ocean which are not in hydrostatic balance (Marshall et al., 1997). CROCO and ROMS_AGRIF have long been applied to solve problems at coastal- or meso-scale ocean problems, such as coupled biogeochemical simulations and submesoscale and river plume simulations, where applying a resolution of more than 1 kilometer is standard. In this paper, CROCO NBQ is used at meter-scale resolution within a total domain size and depth of 100m to 300m. In ocean model simulations, the turbulence tends to moderate with increasing depth. Figure 1 shows that the resulting water mean velocity as simulated by CROCO is sheared as depth increases, the degree to which this occurs results from the activity and momentum transported vertically by turbulence.

The addition of a non-hydrostatic solver is a rare feature to incorporate into a coastal model such as CROCO. From basic fluid mechanics scalings for common oceanographic problems (e.g., McWilliams, 1985), the dimensionless vertical momentum equation has the aspect ratio and Froude number (ratio of vertical shear to buoyancy frequency) as the key parameters determining if hydrostasy is appropriate.

$$\underbrace{\frac{H^2}{L^2}}_{\text{aspect}^2} \underbrace{\frac{V^2}{N^2 H^2}}_{\text{Froude}^2} \frac{Dw}{Dt} = -\frac{\partial \phi}{\partial z} + b \tag{1}$$

In a non-hydrostatic balance, the aspect ratio approaches 1 and the stratification is not stronger than the shear, so both horizontal and vertical accelerations are important, and the resulting turbulent motions are nearly isotropic.

$$\text{Hydrostatic if: } \frac{H}{L} \ll 1, \qquad \text{Non-hydrostatic if: } \frac{H}{L} \sim 1 \text{ and } \frac{V}{NH} \sim 1 \tag{2}$$

Generally, non-hydrostatic ocean modelling is taken on in models that employ the Boussinesq approximation, which result at leading order in incompressible velocities (Marshall et al., 1997). In CROCO, the implementation of non-hydrostatic physics



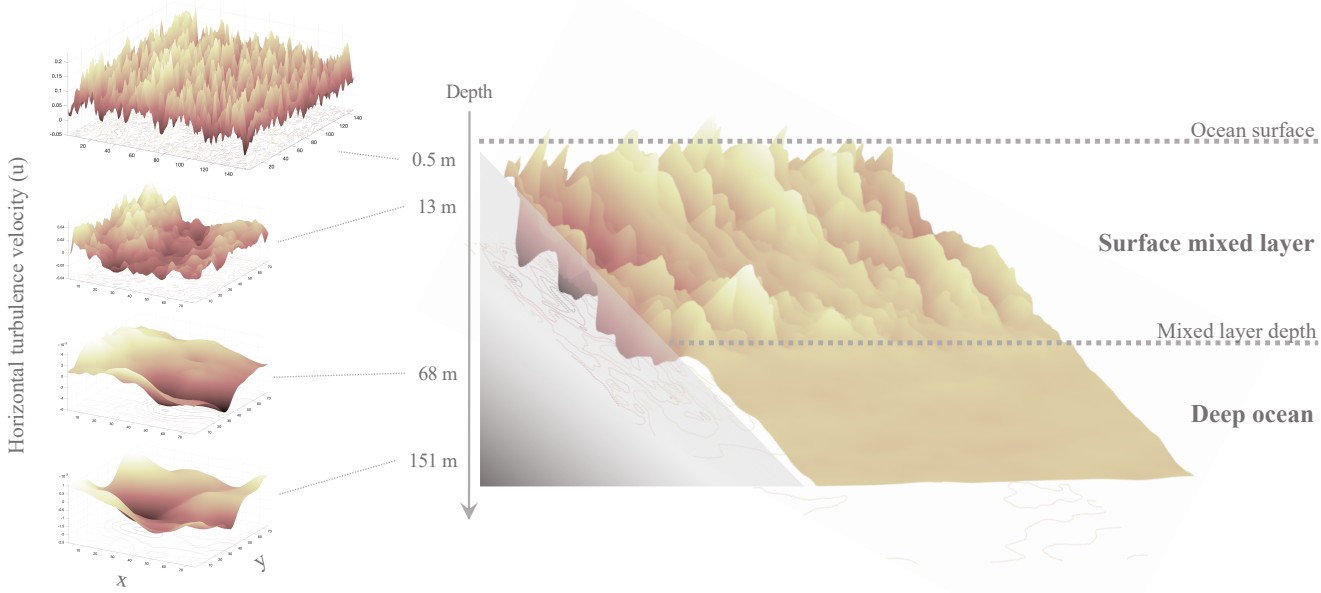

**Figure 1.** The snapshot of the water velocity in a certain direction simulated by CROCO changes with the increase of depth, illustrating the turbulent behavior of CROCO model simulation.

takes advantage of a simplified numerical implementation when compressibility is included. The degree of compressibility can be varied by changing the sound speed in the model, but it cannot be infinite (i.e., incompressible). Indeed, it is desirable to

55 have a sound speed that is slower than in reality, because this reduces the time-step requirements to resolve these waves. A key test of the CROCO system here is to see how slow the sound speed can be made before it begins to affect the accuracy of the results through excessive compressibility.

In order to test the accuracy and computational efficiency of CROCO, an idealized ocean setting is applied as a benchmark, where the proven NCAR Large-Eddy-Simulation (LES) model can be used to evaluate the performance of CROCO. The setting

is doubly-periodic, horizontally-homogeneous turbulence forced with winds and/or convective cooling at the surface following the class of simulations developed for study of entrainment (Li and Fox-Kemper, 2017) and anisotropy (Li and Fox-Kemper, 2020) modeling. NCAR-LES (Moeng, 1984; Sullivan et al., 1994), PALM (Raasch and Schröter, 2001) and Oceananigans (Ramadhan et al., 2020) are branches of the LES model family which are also compared in preliminary testing to see how much those models differ. These LES models are more similar to one another than NCAR-LES and CROCO (they are all

Boussinesq, non-hydrostatic models), but they still differ in capabilities, numerics, code language, and subgrid schemes. The purpose of comparing these three LES models is to demonstrate the level of agreement among "standard" LES models including the NCAR-LES model, which can serve as a guide in the NCAR-LES versus CROCO comparisons. The inter-model spread of the three LES models provides a measure of the level of uncertainty due to SGS parameterizations, numerical schemes, etc. Therefore, in the subsequent analyses with CROCO, the NCAR-LES model is the focus of comparison.





In this paper, the comparisons between CROCO and NCAR-LES are divided into two major aspects: model prediction accuracy (Section 2) and computational efficiency (Section 3). The descriptions of these three LES models are presented in Section 2.5. In the accuracy and LES comparisons, essential turbulence statistics form the basis, and the results include the effects in CROCO of varying the explicit SGS parameterization, the second viscosity, the speed of sound, and the time step. In the efficiency comparison, the computing time for each time step is recorded to measure the model efficiency, and the factors which limit the time step in each model are discussed. Strong and weak scaling are examined in simulations set for different problem sizes and workload per processor, respectively. The impacts of varying the MPI parallelization of CROCO and 2D decomposition of NCAR-LES as well as the settings of the Cheyenne supercomputer are discussed.

## 2  Turbulence statistics accuracy comparison

In this section, we compare the turbulence statistics simulated by the NCAR boundary-layer LES model (Moeng, 1984; Sullivan et al., 1994; Sullivan and Patton, 2011) and the Coastal and Regional Ocean Community (CROCO) non-Boussinesq (NBQ) model (Auclair et al., 2018; Marchesiello et al., 2021). In addition, we test the sensitivity of the turbulence statistics to certain constants specific to the CROCO NBQ model.

All simulations in this section use the following configuration. The grid has 256 uniformly-spaced points in each direction (including the NCAR-LES pseudospectral collocation grid). The domain size is $320\,\mathrm{m} \times 320\,\mathrm{m}$ horizontally and $163.84\,\mathrm{m}$ vertically. The horizontal resolution $\Delta x = \Delta y$ is $1.25\,\mathrm{m}$, and the vertical resolution $\Delta z$ is $0.64\,\mathrm{m}$. The vertical Coriolis parameter $f$ is $1.028 \times 10^{-4}\,\mathrm{s}^{-1}$, and the horizontal Coriolis parameter is set to zero. The density $\rho$ is given by a linear equation of state without salinity: namely, $\rho = \rho_0 + \rho_0 \beta_T (\theta_0 - \theta)$ with the reference density $\rho_0 = 1000\,\mathrm{kg\,m^{-3}}$, reference temperature $\theta_0 = 13.554\,^\circ\mathrm{C}$, the thermal expansion coefficient $\beta_T = 2 \times 10^{-4}\,^\circ\mathrm{C}^{-1}$, and potential temperature $\theta$. Initially, there is a mixed layer having $\theta = 14\,^\circ\mathrm{C}^{-1}$ above $z \geq -42\,\mathrm{m}$, and below that depth the temperature linearly decreases to $12.8\,^\circ\mathrm{C}^{-1}$ at $z = 163.84\,\mathrm{m}$, providing a nearly uniform buoyancy frequency below the mixed layer. The bottom boundary uses a rigid free-slip surface and no-flux conditions. At the upper-boundary, uniform wind stress in the $x$-direction and uniform surface heat flux $Q_*$ are applied where the upper-boundary temperature flux is given by $Q_*/(\rho_0 c_p)$ with specific heat capacity $c_p = 3985\,\mathrm{J\,kg^{-1}\,^\circ C^{-1}}$. The gravitational acceleration $g$ is $9.81\,\mathrm{m\,s^{-1}}$. During the initial spin-up period, the wind stress and the surface heat flux increase to their full values over 51 minutes (5% of the inertial period). After this period, they stay constant. Four combinations of the water-side friction velocity $U_*$ and the surface heat flux $Q_*$ are considered: namely, $(U_*, Q_*) = (0.006\,\mathrm{m\,s^{-1}}, 5\,\mathrm{W\,m^{-2}})$, $(0.006\,\mathrm{m\,s^{-1}}, 50\,\mathrm{W\,m^{-2}})$, $(0.012\,\mathrm{m\,s^{-1}}, 5\,\mathrm{W\,m^{-2}})$, and $(0.012\,\mathrm{m\,s^{-1}}, 50\,\mathrm{W\,m^{-2}})$.

The NCAR LES model uses a two-part SGS eddy-viscosity model of Sullivan et al. (1994) designed to improve the LES accuracy in comparison to similarity theory (Monin and Obukhov, 1954) near the surface at $z = 0\,\mathrm{m}$. Their SGS model constants $C_k$ and $C_\epsilon$ in their equations 4 and 11 are 0.1 and 0.93, respectively. We configure their SGS model such that it reduces to a simpler form (their equation 1) below $z = -21\,\mathrm{m}$. With rough approximations, this simpler model can be related to the Smagorinsky (1963) model with a relatively large value of the corresponding Smagorinsky constant $C_s = 0.18$ (their equation 14). The NCAR LES uses the pseudo-spectral method (Fox and Orszag, 1973) for the horizontal derivatives and second-order



centered finite-differences for the vertical derivatives (Moeng, 1984). The resolved vertical temperature flux is determined using a second-order near monotonic scheme (Beets and Koren, 1996). The higher third of wavenumbers are zeroed out to

remove aliasing of unresolved scales (Orszag, 1971). The time stepping utilizes a third-order Runge-Kutta scheme (Sullivan et al., 1996). More information is given in the model description papers (Moeng, 1984; Sullivan et al., 1994; Sullivan and Patton, 2011).

The CROCO NBQ model offers several options for the SGS parameterizations. In this paper, we consider two options: namely, the use of only numerical diffusion and the SGS model of Lilly (1962). The former avoids adding any explicit SGS

terms and implicitly relies only on numerical diffusion. Here, the WENO5-Z improved version of the 5th-order weighted essentially nonoscillatory scheme (Borges et al., 2008) is used for all advection terms (see Auclair et al., 2018; Marchesiello et al., 2021, for more information). Unless explicitly mentioned otherwise, the CROCO runs shown here use the numerical-only option for the SGS parametrization because we are interested in understanding the performance of (unavoidable) numerical diffusion before adding explicit SGS terms (and associated parameters) which make the model behavior more complex. We

test the explicit SGS effect only briefly in section 2.2.

## 2.1    NCAR LES model vs CROCO NBQ model

Here, we compare the NCAR LES model with the CROCO NBQ model. As we will see shortly, the results show that these two models produce very similar boundary-layer flows, with differences comparable to those among the different LES (Section 2.5).

The CROCO model uses a time-splitting method and uses two different time steps for the so-called fast and slow modes.

In this subsection, all of the CROCO runs use a slow-mode time-step of $0.5\,\mathrm{s}$ and a fast-mode time-step of $0.019\,\mathrm{s}$. We tested many different time steps, and these values are clost to the largest stable values for the configuration used. To match the slow-mode time step, the NCAR model runs in this section use a time-step of $0.5\,\mathrm{s}$ as well. However, note that the NCAR model can be run with a much larger time step and it has the capability of adjusting its time-step based on an embedded Runge-Kutta multiple-order approach; namely, the Courant-Friedrichs-Lewy (CFL) time-step which the NCAR model finds when running

with adjustable time-stepping is about $7\,\mathrm{s}$ for the run with $U_* = 0.006\,\mathrm{m\,s^{-1}}$ and about $3\,\mathrm{s}$ for the runs with $U_* = 0.012\,\mathrm{m\,s^{-1}}$. Thus, when used with this reduced time-step the NCAR model is roughly fourteen times slower.

The CROCO NBQ model has two constants related to the fast mode: namely, the speed of sound $c_s$ and the second viscosity (also called bulk viscosity, volume viscosity, or dilatational viscosity) $\lambda$. Because we are not interested in sound waves, we may use an unphysically-small value of $c_s$ and an unphysically-large value of $\lambda$ to relax the sound-related CFL constraint by

slowing and damping these waves, respectively. In this subsection, we use $c_s = 3\,\mathrm{m\,s^{-1}}$ and $\lambda = 1\,\mathrm{kg\,s^{-1}\,m^{-1}}$, which are about 500 times slower and 400 times more viscous than in seawater. As shown in sections 2.4 and 2.3, the unphysical values of these constants affect turbulence statistics negligibly.



Figures 2 and 3 show the vertical profiles of various flow properties.[1] Hereafter, we use the following symbols: the horizontal average $\overline{\phi}$ and the turbulent fluctuation $\phi' \equiv \phi - \overline{\phi}$ for any quantity $\phi$, the buoyancy $b \equiv -g\rho/\rho_0$, the buoyancy frequency

$N^2 \equiv \partial \overline{b}/\partial z$, and the horizontally-averaged depth $z_p$ of the mixed-layer base defined as the $z$-coordinate of the $N^2$ maximum.

### 2.1.1 Scaling of turbulent properties

To understand these figures, let us first explain the nondimensionalization used. Figures 2a, 3a, 3b, 3c, and 3f show quantities related to the turbulent kinetic energy (TKE) and the TKE shear production such as the mean shear and a Reynolds stress component. These quantities are largely governed by the energy input to the water rather than the wind stress or surface heat

flux. Therefore, we introduce a characteristic scale $E_*$ of the surface energy flux:

$$E_* \equiv U_*^2 \overline{u}_0 + B_* |z_p| \tag{3}$$

where $\overline{u}_0(t) \equiv \overline{u(x,y,z=0,t)}$ is the surface current in the wind-stress direction, and $B_* \equiv g\beta_T Q_*/(\rho_0 c_p)$ is the surface buoyancy flux. The first term on the r.h.s. is the flux of the work done by the wind stress, and the second term is a rough approximation of the flux of available potential energy.[2] For ease of notation, we use an energy-flux-based velocity scale

$$U_E \equiv E_*^{\frac{1}{3}}. \tag{4}$$

While $\overline{v}$ and $\overline{u'w'}$ are also related to the TKE shear production, they are largely constrained by other factors. Therefore, we use other scalings to nondimensionalize them. Namely, figure 2b uses the vertically-averaged Ekman transport velocity $U_*^2/(f|z_p|)$ because $\overline{v}$ is roughly constrained by the Ekman balance. Figure 3e uses the wind stress $U_*^2$ because $\overline{u'w'}$ is constrained by the wind stress.

In figure 2d, we use a stratification scale $\Gamma_N$ pertinent to pycnocline entrainment where

$$\Gamma_N \equiv \frac{2E_b^{\frac{2}{3}}}{\Delta_e (z_w - z_p)}, \tag{5}$$

and $\Delta_e$ is a length scale[3], and

$$z_w \equiv -\frac{U_E}{4.5f} \tag{6}$$

is a rough depth scale of the wind-driven boundary layer[4], and

$$E_b \equiv U_*^2 \overline{u}_0 e^{-\frac{z_p}{z_w}} + B_* |z_p| \tag{7}$$

---

[1]Each profile is an average of 21 samples taken every one-fortieth (about 25 minutes) of the inertial period during $t = 4.7$ to 13.6 hours. At each given time, the normalized profiles are computed using the characteristic scales at that time. Then, the final profiles are made by averaging these normalized profiles. The time window is kept short, about 9 hours, because the simulated flow is not in a statistically steady state due to mixed-layer deepening and entrainment. In all simulations, the boundary-layer thickness reaches the initial mixed-layer thickness within 4 hours from $t = 0$ s when the flow has no motion.

[2]Here, for notational simplicity, we use a positive value when energy is coming into the water.

[3]The length scale $\Delta_e$ is independent of the flow. Therefore, an arbitrary value may be used. Here we arbitrarily use $\Delta_e = 1$ m.

[4]The factor 4.5 is an empirical nondimensional coefficient. Equation (6) is related to the standard thickness of the Ekman layer derived assuming a constant vertical eddy viscosity. Here, however, we relate the wind-driven boundary-layer thickness to the surface energy flux because the eddy viscosity does not have to be vertically uniform but is still roughly related to the surface energy flux.





is a rough scale of the energy flux at $z_p$ causing pycnocline entrainment.[5] Unlike the available potential energy input, the wind energy input is largely dissipated near the surface and is not directly used for pycnocline entrainment. Therefore, (7) assumes an exponential decay of the wind energy available to pycnocline entrainment. Note that, for a pycnocline buoyancy frequency $N_p^2$, $(z_w - z_p)\Delta_e N_p^2/2$ is the energy necessary to mix $\Delta_e$ thickness of the pycnocline water with the adjacent mixed-layer water located between $z_w$ and $z_p$ where mostly the convective turbulence has to entrain the pycnocline water and lift it up to the Ekman-layer bottom $z_w$ (where a larger amount of wind energy is available to the mixing above). Therefore, the normalized buoyancy frequency in figure 2d indicates how strong the pycnocline stratification is relative to the energy input available for the pycnocline entrainment.

In figure 2e, we use a two-part buoyancy flux scale

$$\Gamma_{b'w'} \equiv \max\left(1 - \frac{z}{z_p}, 0\right) B_* + \min\left(\frac{z}{z_p}, 1\right) E_b^{\frac{2}{3}} \sqrt{N_p^2} \times 4 \times 10^{-3} \tag{8}$$

where the first term is the scale relevant near the surface and the second term is the scale relevant near the boundary-layer bottom. The nondimensional constant $4 \times 10^{-3}$ in the second term is used only to make the normalized value at $z_p$ close to -1.

Figure 2f uses the energy-flux-based scale for $w'w'w'$ but modified with a nondimensional function $\phi_s$ as

$$\Gamma_{w'w'w'} \equiv \phi_s U_E^3 \tag{9}$$

because $w'w'w'$ is very sensitive to the turbulence structure. When $(U_*, Q_*) = (0.006\,\mathrm{m\,s^{-1}}, 50\,\mathrm{W\,m^{-2}})$, the turbulence develops distinct convective rolls spanning the whole boundary-layer depth while in other cases convective rolls are much weaker and the turbulence structures in the upper part of the boundary layer are more similar to the pure wind-driven turbulence–which mainly consists of smaller-scale and more-disturbed tilted-vortexes–and the turbulence structures in the lower part of the boundary layer are similar to pure convective plumes. Convective rolls utilize both wind energy and available potential energy constructively and channel these energies into bands of strong $w'$. In contrast, the turbulence in the other cases uses wind energy to mix the water in the upper part of the boundary layer and thereby partially distracts the available potential energy coming in from the surface. As a result, $\overline{w'w'w'}$ due to convective rolls is much stronger. Therefore, to make the order of the normalized values similar, we use $\phi_s = 5$ when $(U_*, Q_*) = (0.006\,\mathrm{m\,s^{-1}}, 50\,\mathrm{W\,m^{-2}})$ and $\phi_s = 1$ otherwise.

Figure 3d shows $\overline{b'b'}$ near $z_p$. It is dominated by internal waves and isopycnal deformation due to the boundary-layer turbulence reaching $z_p$. The nondimensionalization is done relative to the stratification and the energy input to these processes: namely,

$$\Gamma_{b'b'} \equiv E_b^{\frac{2}{3}} N^2. \tag{10}$$

### 2.1.2 Comparison of results

Figures 2a, 2b, and 2c show that the simulated mean flows are very similar. The only somewhat notable differences are 1) that the CROCO surface velocity tends to be slightly higher, 2) that the CROCO surface temperature tends to be slightly lower,

---

[5]When the wind energy mixes the surface water very well and thereby siginifically distract the available potential energy due to the surface cooling, it may be more appropriate to use $B_*|(z_w - z_p)|$ instead in the second term.





and 3) that the CROCO pycnocline entrainment is weaker. The weaker entrainment in CROCO can be seen more clearly in the comparison of the deepening mixed layers in Figure 4.

The CROCO runs produced weaker mixed-layer deepening although Figure 2d shows that CROCO runs had either a similar or greater amount of energy flux reaching the mixed-layer base.[6] Furthermore, despite the slower mixed-layer deepening
in the CROCO runs, CROCO tends to have a slightly stronger resolved buoyancy flux at the mixed-layer base (figure 2e). This implies that the NCAR model's faster entrainment occurs because NCAR model's explicit SGS diffusion is larger than CROCO's implicit (numerical-only) SGS diffusion. Note that the NCAR model also has only second-order advection in the vertical with upwinding, so even though it is centered it may have higher-order diffusion and dispersion effects, while CROCO has fifth-order advection with implicit diffusion entering only at the highest orders and limited third-order dispersion errors.
This point is reiterated in section 2.2 where we add explicit SGS diffusion terms to a CROCO run.

Figures 2e, 2f, 3a, 3b, 3c, 3e, and 3f show that the resolved turbulence statistics are overall very similar. Note that a difference of up to about 10 % should be considered negligible for the domain size used and the time window lengths used for averaging. Experimentation by varying time-steps (not shown) gives this level of difference, reflecting that different realizations of instantaneous chaotic turbulent flow that do not altering the turbulence statistics can differ by this amount. This order of difference
is likewise justified by the comparison among the LES in Section 2.5, which approach 10% differences in many of the same variables even though the averaging in those LES comparison figures is over an entire inertial period. Especially, the profiles of $\overline{w'w'w'}$, $\overline{v'w'}$, and $\overline{b'b'}$ fluctuate strongly and require significant averaging to obtain a well-sampled profile. However, near the surface where the turbulence structures tend to be small, the statistics are more robust even for these quantities. Thus, the resolved turbulence quantities near the surface tend to be robustly stronger for the CROCO runs. This stronger resolved turbu-
lence is closely related to the difference in the SGS parameterization, which becomes significant near the surface. Generally, a stronger SGS diffusion tends to weaken the resolved turbulence. Therefore, the result here suggest that CROCO's numerical diffusion is weaker than the explicit SGS diffusion of the NCAR model. As shown in section 2.2, the difference in the resolved turbulence quantities significantly reduces when the CROCO model uses an explicit SGS diffusion additionally to the (unavoidable) numerical diffusion.

Figures 3c and 3d show that the variances of the resolved $w$ and $b$ in the stratified part of the water ($z/|z_p| \lesssim -0.9$) tend to be larger with the NCAR model. This is partially due to the slightly smaller $U_E$ and $E_b$ in the NCAR runs. However, this tendency persists in the dimensional variances as well. Contrary to these variances, the resolved buoyancy flux (figure 2e) at the same depths tends to be less with the NCAR model. Therefore, the NCAR runs have stronger internal waves (who have no buoyancy flux when they are not growing nor decaying) and less resolved turbulent mixing. It is not clear why this is the case,
but one hypothesis was that these waves are more easily supported by the horizontal pseudospectral numerics of the NCAR LES, and another hypothesis is that the fifth-order WENO scheme in CROCO is damping these waves. Further experimentation

---

[6]That is, the normalized buoyancy frequency of the pycnocline tends to be smaller for the CROCO runs while the dimensional $N^2$ of the pycnocline is the same for both NCAR and CROCO runs. This is a result of a slightly larger $\overline{u}_0$ in the CROCO runs, which leads to a larger $U_E$, a deeper $z_w$, and a smaller $z_w - z_p$, a larger $E_b$, and a larger $\Gamma_N$.





with different numerics in CROCO is possible, but is beyond the scope of this comparison paper. However, no similar effect is seen when comparing the different LES schemes in Figures 14-15.

To further investigate this difference, the spectra of 1D discrete FFT modes and the circularly-integrated 2D energy spectra of $u'$, $v'$, $w'$, and $b'$ are shown in figures 5 and 6. These figures are made using the data taken from special runs having a larger horizontal domain size of $640\,\mathrm{m} \times 640\,\mathrm{m}$ to have more wavenumbers and for better statistics, and the results are very similar to the baseline domain size of $320\,\mathrm{m} \times 320\,\mathrm{m}$. These spectra are taken from three different regions: namely, the mixed-layer interior ($-32\,\mathrm{m} < z < -6\,\mathrm{m}$), the entrainment layer ($-60\,\mathrm{m} < z < -38\,\mathrm{m}$), and the pycnocline interior ($-132\,\mathrm{m} < z < -70\,\mathrm{m}$). Overall, the NCAR and CROCO simulations tend to differ at the spectral heads and tails. The difference in the

high-wavenumber tail is likely due to the dealiasing truncation in the NCAR runs which is not likely to resemble the high-wavenumber numerical diffusion in the CROCO approach. Note that this difference occurs over roughly the upper third of wavenumbers where the dealiasing is applied. We show only the spectra from the case with $(U_*, Q_*) = (0.012\,\mathrm{m\,s^{-1}}, 5\,\mathrm{W\,m^{-2}})$ because the differences between the NCAR and CROCO simulations have similar tendency for all other cases.





**Figure 2.** Comparison between the NCAR LES model (solid) and the CROCO NBQ model (dashed). The line color indicates the surface forcing as shown in the legend.





**Figure 3.** Comparison between the NCAR LES model (solid) and the CROCO NBQ model (dashed). The line color indicates the surface forcing as shown in the legend.





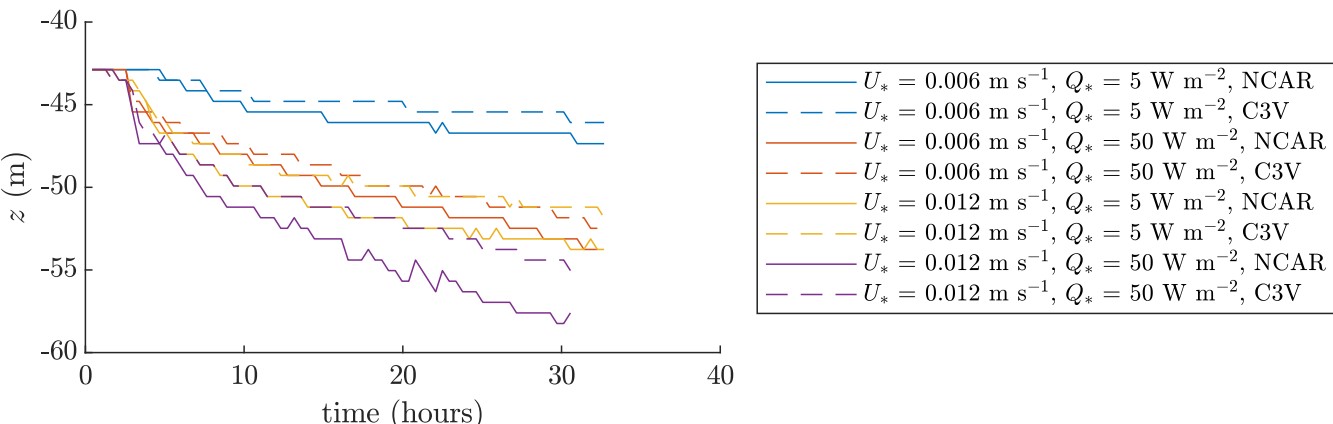

**Figure 4.** Time series of the mixed-layer-base depth $z_p$. C3V in the legend refers to the CROCO NBQ run with the sound speed $c_s = 3\,\mathrm{m\,s^{-1}}$ and the second viscosity $\lambda = 1\,\mathrm{kg\,s^{-1}\,m^{-1}}$. The difference in the mixed-layer deepening occurs mainly because NCAR's explicit SGS diffusion is larger than CROCO's implicit SGS (that is, only numerical) diffusion.







**Figure 5.** Comparison of the 1D discrete FFT spectra with $(U_*, Q_*) = (0.012 \, \mathrm{m \, s^{-1}}, 5 \, \mathrm{W \, m^{-2}})$. Each spectrum is smoothed by averaging over the vertical range shown in each title as well as averaging over 21 hours and each horizontal direction.





**Figure 6.** Comparison of the 2D spectra averaged in circular rings at constant horizontal wavenumber magnitude from the runs with $(U_*, Q_*) = (0.012\,\mathrm{m\,s^{-1}}, 5\,\mathrm{W\,m^{-2}})$. Each spectrum is smoothed by averaging over the vertical range shown in each title as well as averaging over 21 hours.



## 2.2 The effect of the explicit SGS parameterization

This subsection shows how explicit SGS diffusion terms affect the results in subsection 2.1. For this, we focus on the case with $(U_*, Q_*) = (0.012\,\mathrm{m\,s}^{-1}, 50\,\mathrm{W\,m}^{-2})$ because this case has the largest difference in the mixed-layer deepening, which is the most significant difference observed in the previous subsection.

Here, the CROCO NBQ run uses a modified version of the SGS parameterization by Lilly (1962). Namely,

$$\tau_{ih} = \nu_H \left( \frac{\partial u_i}{\partial x_h} + \frac{\partial u_h}{\partial x_i} \right), \tag{11}$$

$$\tau_{i3} = \nu_V \left( \frac{\partial u_i}{\partial z} + \frac{\partial w}{\partial x_i} \right), \tag{12}$$

$$\tau_{\theta h} = \Pr \nu_H \frac{\partial \theta}{\partial x_h}, \tag{13}$$

$$\tau_{\theta z} = \Pr \nu_V \frac{\partial \theta}{\partial z}, \tag{14}$$

where

$$S_{ij} = \frac{1}{2} \left( \frac{\partial u_i}{\partial x_j} + \frac{\partial u_j}{\partial x_i} \right), \tag{15}$$

$$D = \sqrt{2 S_{ij} S_{ij}}, \tag{16}$$

$$\nu_H = C_s^2 \Delta x \Delta y D \sqrt{\max\left( 0, \, 1 - \frac{N^2/D^2}{C_R} \right)}, \tag{17}$$

$$\nu_V = C_s^2 \Delta z \Delta z D \sqrt{\max\left( 0, \, 1 - \frac{N^2/D^2}{C_R} \right)}, \tag{18}$$

and the indexes are $h = 1, 2$, $i = 1, 2, 3$, and $j = 1, 2, 3$, and the summation convention is used, and the model parameters are the Smagorinsky constant $C_s$, Prandtl number $\Pr$, and a mixing-threshold constant $C_R$. The SGS terms become zero 245 when a Richardson-like number $N^2/D^2$ exceeds $C_R$. As mentioned in the introduction of section 2, the NCAR model's SGS parameterization below $z = -21\,\mathrm{m}$ is roughly relatable to the Smagorinsky model with $C_s = 0.18$. Therefore, we test $C_s = 0.17$ and $0.2$ with CROCO. These values of $C_s$ together with a large value of $\Pr$ produce the mixed-layer deepening comparable to the NCAR model run as shown in Figure 7 where the mixed-layer deepening with $(C_s, C_R, \Pr) = (0.17, 0.25, 3)$ and $(0.2, 1, 4)$ are shown. Note also that the net entrainment in the CROCO implicit plus explicit diffusion cases (C3VS 250 and C3VS2) is greater that the implicit-only diffusion, which is important to verify as occasionally net effects can in fact become larger under implicit-only diffusion if the gradients sharpen in response (Bachman et al., 2017). The results in Figure 7 demonstrate that the difference in the entrainment and mixed-layer deepening seen in the previous subsection is due primarily to the SGS parameterization always present in the NCAR LES, and the numerical-only diffusion of the CROCO runs is less than the combined numerical plus explicit diffusion of the NCAR model and the C3VS cases.

The previous subsection also showed that the resolved turbulence quantities near the surface tend to be larger with the CROCO model without an explicit SGS parameterization. This difference also significantly reduces with the addition of the





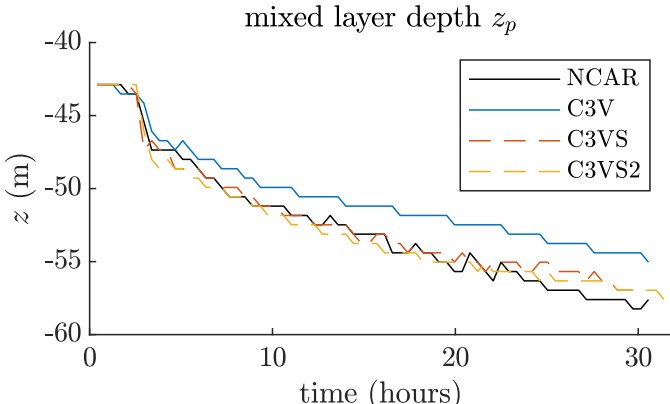

**Figure 7.** Time series of the mixed-layer-base depth $z_p$. The CROCO NBQ runs (C3V, C3VS, C3VS2 in the legend) use the sound speed $c_s = 3\,\mathrm{m\,s^{-1}}$ and the second viscosity $\lambda = 1\,\mathrm{kg\,s^{-1}\,m^{-1}}$. C3V uses only numerical diffusion. C3VS and C3VS2 use an explicit SGS parameterization (11)-(18) with $(C_s, C_R, \mathrm{Pr}) = (0.17, 0.25, 3)$ and $(0.2, 1, 4)$, respectively.

explicit SGS parameterization as shown in Figures 8 and 9.[7] A stronger near-surface diffusion weakens the resolved turbulence. There are some small remaining differences, but they are expected because different explicit SGS parameterizations are used in the NCAR and CROCO models.

In summary, the NCAR results and the CROCO results are overall very comparable. There are some minor differences, but most of them are due to the different SGS parameterization. The only notable difference that may not be attributable to the SGS parameterization difference is that the NCAR model runs tend to produce more internal waves in the stratified part.

---

[7]Each profile is an average of 21 samples taken every one-fortieth of the inertial period during $t = 4.7$ to 13.6 hours.







**Figure 8.** Comparison between the NCAR run (solid) and the C3VS CROCO run (dashed) including explicit SGS dissipation with $(C_s, C_R, \mathrm{Pr}) = (0.17, 0.25, 3)$. Compare to the purple lines in Figure 2 which show the same forcing but the CROCO C3V case with only implicit dissipation.



**Figure 9.** Comparison between the NCAR run (solid) and the C3VS CROCO run (dashed) including explicit SGS dissipation with $(C_s, C_R, \mathrm{Pr}) = (0.17, 0.25, 3)$. Compare to purple lines in Figure 3 which show the same forcing but with CROCO C3V case with only implicit dissipation .



## 2.3 The effect of the second viscosity parameter

For the CROCO NBQ model runs, an unphysically-large value of the second viscosity $\lambda$ may be used to aggressively dissipate
(near-grid-scale) pseudo-acoustic waves and stabilize the simulation. Therefore, here we test whether an unphysically-large
value of $\lambda$ affects the turbulence statistics. The results show that the turbulence statistics are not significantly affected.

We present two types of CROCO runs having the speed-of-sound parameter $c_s = 202\,\mathrm{m\,s^{-1}}$. One type (referred to as C202)
uses $\lambda = 0.01\,\mathrm{kg\,s^{-1}\,m^{-1}}$, and the other type (referred to as C202V) uses $\lambda = 19\,\mathrm{kg\,s^{-1}\,m^{-1}}$ for $(U_*, Q_*) = (0.006\,\mathrm{m\,s^{-1}}, 50\,\mathrm{W\,m^{-2}})$
and $\lambda = 18\,\mathrm{kg\,s^{-1}\,m^{-1}}$ for all other values of $(U_*, Q_*)$. Figures 10 and 11[8] show the flow statistics from C202 and C202V are
essentially unchanged by the higher viscosity.

By increasing $\lambda$, the additional viscosity does have the effect of stabilizing marginal numerical instabilities so that the
optimal slow-mode time step increases from $0.15\,\mathrm{s}$ to $0.2\,\mathrm{s}$ for the case with $(U_*, Q_*) = (0.006\,\mathrm{m\,s^{-1}}, 5\,\mathrm{W\,m^{-2}})$, and from
$0.04\,\mathrm{s}$ to $0.08\,\mathrm{s}$ for the cases with $(U_*, Q_*) = (0.012\,\mathrm{m\,s^{-1}}, 5\,\mathrm{W\,m^{-2}})$ and $(0.012\,\mathrm{m\,s^{-1}}, 50\,\mathrm{W\,m^{-2}})$. However, for the case
with $(U_*, Q_*) = (0.006\,\mathrm{m\,s^{-1}}, 50\,\mathrm{W\,m^{-2}})$, increasing $\lambda$ does not lead to an increase of the slow-mode time, which stays at
$0.25\,\mathrm{s}$. The optimal fast-mode time step is unaffected by $\lambda$ and is about $0.0038\,\mathrm{s}$ for all values of $(U_*, Q_*)$. Therefore, increasing
damping using $\lambda$ speeds up the simulations only slightly.

---

[8]The initial conditions for the C202 and C202V runs are prepared by simulating the boundary layers for 4 hours using $c_s = 3\,\mathrm{m\,s^{-1}}$ and $\lambda = 1\,\mathrm{kg\,s^{-1}\,m^{-1}}$
from a quiescent state. The boundary layers fully develop during this time. At the time of the initial conditions, we reset $t = 0\,\mathrm{s}$, $c_s = 202\,\mathrm{m\,s^{-1}}$, and $\lambda$
to the new values. Every profile from $(U_*, Q_*) = (0.006\,\mathrm{m\,s^{-1}}, 5\,\mathrm{W\,m^{-2}})$, $(0.006\,\mathrm{m\,s^{-1}}, 50\,\mathrm{W\,m^{-2}})$, $(0.012\,\mathrm{m\,s^{-1}}, 5\,\mathrm{W\,m^{-2}})$, and $(0.012\,\mathrm{m\,s^{-1}},$
$50\,\mathrm{W\,m^{-2}})$ is an average of the samples taken every one-fortieth of the inertial period during $t = $ 4-7, 2.5-5.5, 2-11, and 2-11 hours, respectively.





**Figure 10.** The effect of the second viscosity $\lambda$. The C202 runs (solid) use $\lambda = 0.01\,\mathrm{kg\,s^{-1}\,m^{-1}}$, and the C202V runs (dashed) use $\lambda = 18\,\mathrm{kg\,s^{-1}\,m^{-1}}$ to $19\,\mathrm{kg\,s^{-1}\,m^{-1}}$.





**Figure 11.** The effect of the second viscosity $\lambda$. The C202 runs (solid) use $\lambda = 0.01\,\mathrm{kg\,s^{-1}\,m^{-1}}$, and the C202V runs (dashed) use $\lambda = 18\,\mathrm{kg\,s^{-1}\,m^{-1}}$ to $19\,\mathrm{kg\,s^{-1}\,m^{-1}}$.



## 2.4 Sensitivity to the speed-of-sound parameter

Reducing the speed-of-sound parameter $c_s$ in the CROCO NBQ model allows a larger time step by relaxing the CFL condition related to pseudo-acoustic waves. Here, we study the effect of reducing the value of $c_s$ to a very small value, $c_s = 3\,\mathrm{m\,s^{-1}}$. The results show that the resolved turbulence statistics are largely insensitive to the value of $c_s$. However, it should be noted that $c_s$ should not be smaller than the fastest speed of the process that needs to be properly simulated, for example, the barotropic wave speed in the case of geophysical applications.

Figures 12 and 13[9] compare two types of CROCO runs: one (referred to as C3V) uses $c_s = 3\,\mathrm{m\,s^{-1}}$ and the second viscosity $\lambda = 1\,\mathrm{kg\,s^{-1}\,m^{-1}}$, and the other (referred to as C202) uses $c_s = 202\,\mathrm{m\,s^{-1}}$ and $\lambda = 0.01\,\mathrm{kg\,s^{-1}\,m^{-1}}$. Most profiles in these figures show only small differences that should be considered negligible for the given limited domain size. The only possibly non-negligible difference appears in the internal wave strength seen below $z/|z_p| \approx -0.9$ in Figures 13c and 13d for the cases with $U_* = 0.012\,\mathrm{m\,s^{-1}}$. It is unclear why the intensity of internal waves is sensitive to changing the sound speed. However, note that the differences among the two CROCO simulations in Figure 13 are not as big as the differences between NCAR and CROCO internal wave strength in previous comparisons.

By decreasing $c_s$ together with increasing $\lambda$, the optimal slow-mode and fast-mode time steps increase to $0.5\,\mathrm{s}$ and $0.019\,\mathrm{s}$, respectively, for all C3V runs.[10] Therefore, compared to the C202 runs, C3V runs are more than 5 times faster.

---

[9]Every profile from $(U_*, Q_*) = (0.006\,\mathrm{m\,s^{-1}},\ 5\,\mathrm{W\,m^{-2}})$, $(0.006\,\mathrm{m\,s^{-1}},\ 50\,\mathrm{W\,m^{-2}})$, $(0.012\,\mathrm{m\,s^{-1}},\ 5\,\mathrm{W\,m^{-2}})$, and $(0.012\,\mathrm{m\,s^{-1}},\ 50\,\mathrm{W\,m^{-2}})$ is an average of the samples taken every one-fortieth of the inertial period during a time window of 3, 4.5, 9, and 9 hours starting from 4, 2, 2, and 2 hours, respectively, after the simulations' initial conditions. The initial conditions are made by simulating the boundary layer for 4 hours by C3V. The boundary layer fully develops in 4 hours from a quiescent state.

[10]Decreasing $c_s$ without increasing $\lambda$ makes simulations unstable. Therefore, it is not recommendable.



**Figure 12.** The effect of the speed-of-sound parameter $c_s$. The solid lines are with $c_s = 202\,\mathrm{m\,s}^{-1}$, and the dashed lines are with $c_s = 3\,\mathrm{m\,s}^{-1}$.





**Figure 13.** The effect of the speed-of-sound parameter $c_s$. The solid lines are with $c_s = 202\,\mathrm{m\,s}^{-1}$, and the dashed lines are with $c_s = 3\,\mathrm{m\,s}^{-1}$.





## 2.5 Comparison between LES models

NCAR-LES was developed at NCAR to simulate planetary boundary layer turbulence (Moeng, 1984) and extended to include the effects of ocean surface waves when applied to the ocean surface boundary layer turbulence (McWilliams et al., 1997). The spatial discretization is pseudo-spectral in the horizontal and finite-difference in the vertical. It uses a modified Smagorinsky sub-grid scale (SGS) closure that evolves a prognostic equation for the SGS turbulent kinetic energy (TKE) (Deardorff, 1980; Sullivan et al., 1994).

The Parallelized Large-Eddy Simulation Model (PALM) was developed at Leibniz Universität Hannover (Germany) as a turbulence-resolving LES model for atmospheric and oceanic boundary layer flows, specifically designed to run on massively parallel computer architectures (Raasch and Schröter, 2001; Maronga et al., 2015). It uses a modified version of the Deardorff (1980) SGS parameterization similar to the NCAR-LES. But the spatial discretization is finite-differences in both horizontal and vertical directions. An upwind-biased fifth-order differencing scheme for advection terms in combination with a third-order Runge–Kutta time-stepping scheme is used in PALM (Wicker and Skamarock, 2002).

Both NCAR-LES and PALM have been widely used in simulating atmospheric and oceanic boundary layer turbulence under various idealized and realistic conditions, while Oceananigans is a new fast and friendly software package for numerical simulations of geophysical fluid dynamics developed at the Massachussetts Institute of Technology in the Julia programming language (Ramadhan et al., 2020). Oceananigans uses a spatial discretization that is finite-volume and it can be configured as an LES with various combinations of SGS, advection and time-stepping schemes. For this particular comparison, we are using the anisotropic minimum dissipation closure (Verstappen, 2018) combined with third-order Runge-Kutta time-stepping and fifth-order WENO advection.

Ideally, the differences in the discretization and SGS closure schemes among the three LES models should not affect the horizontally and temporally averaged turbulence statistics for the ocean surface boundary layer problem, as long as the grid cells are small enough to capture the dominant turbulent structures and the model domain is large enough to collect robust statistics. Here we assess to what extent is this assumption valid using two idealized cases: a case dominated by wind driven shear turbulence with $(U_*, Q_*) = (0.012\,\mathrm{m\,s^{-1}}, 5\,\mathrm{W\,m^{-2}})$ and a case dominated by convective turbulence with $(U_*, Q_*) = (0.006\,\mathrm{m\,s^{-1}}, 500\,\mathrm{W\,m^{-2}})$. In both cases, we run PALM and Oceananigans using a consistent domain size and resolution as NCAR-LES.

Figures 14 and 15 compare the vertical profiles of the horizontal mean turbulence statistics averaged over the last inertial period ($\sim$17 hours) for the two idealized cases, respectively. As expected, the three LES models give largely consistent results for the turbulence statistics examined here in the two idealized cases. The most notable differences are confined near the surface or the base of the boundary layer where entrainment is important, which highlights where the differences in SGS closure and numerical schemes have their greatest impact (note that the turbulence statistics shown here are all well-resolved). The seemingly large discrepancies in Figures 14c and 15a,d are due to the normalization and constraints imposed at the surface: the buoyancy flux is small in the wind-driven-shear turbulence dominant case, and the momentum flux is small in the convective turbulence dominant case. Thus, the variables which are most strongly forced at the surface are in closest agreement when



normalized by the surface forcing (Li and Fox-Kemper, 2017; Skitka et al., 2020). Indeed, the simulated momentum flux in the first case (Figure 14d) and buoyancy flux in the second case (Figure 15) show the best agreement among the three LES models. The vertical velocity skewness ($\overline{w'^3}$), cross-wind velocity component ($\overline{v'^2}$), temperature variance ($\overline{t'^2}$), and stratification ($\overline{N^2}$) (panels b, e, and f) are not as strongly constrained by the surface forcing and are subject to more variability among the models, especially at the surface and base of the boundary layer where entrainment occurs.



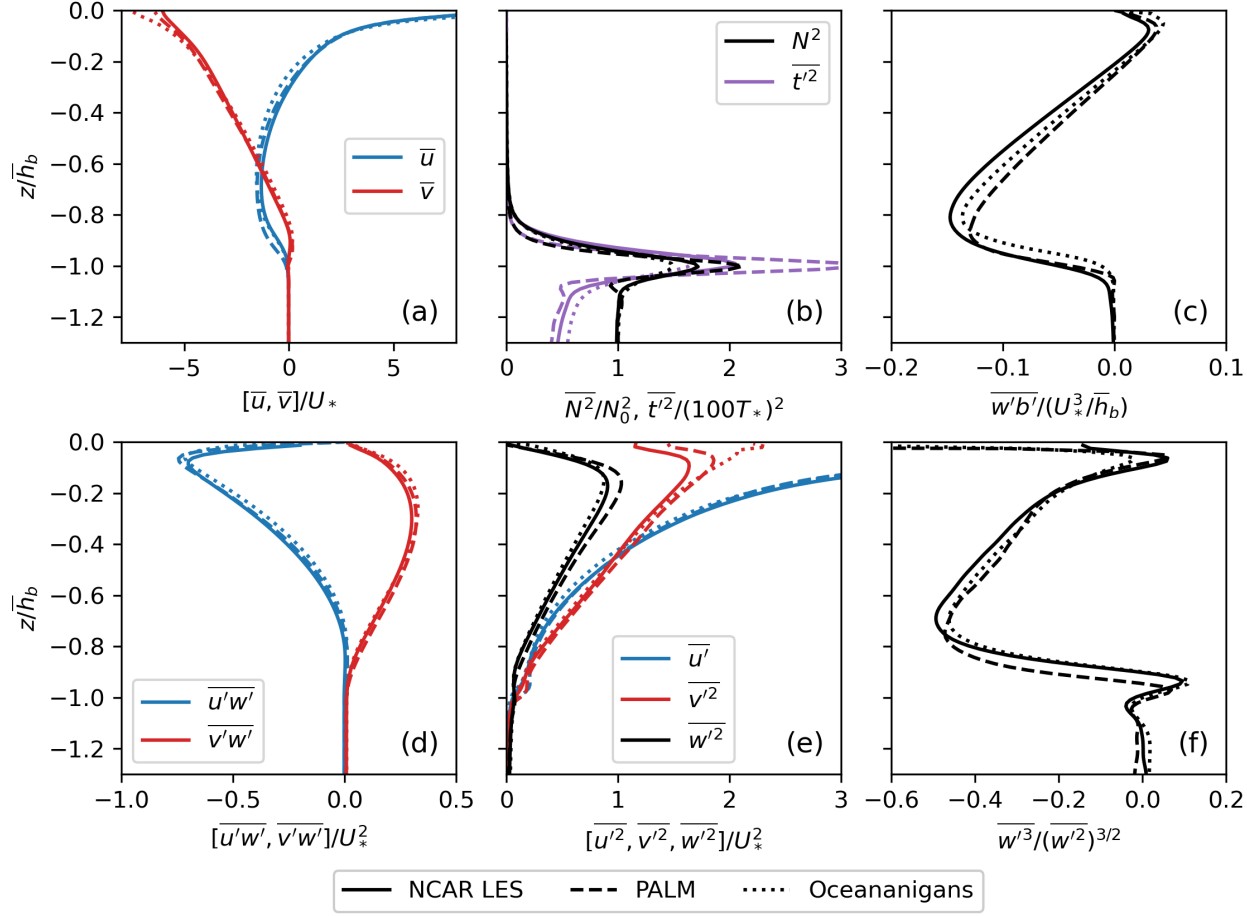

**Figure 14.** A comparison of the horizontally and temporally averaged turbulence statistics among NCAR-LES (solid), PALM (dashed) and Oceananigans (dotted) in a case dominated by wind driven shear turbulence. The normalized turbulence statistics include: (a) horizontal velocity, $\overline{u}, \overline{v}$, normalized by the friction velocity $U_*$; (b) stratification (black), $\overline{N^2}$, normalized by its value below the boundary layer, $N_0^2$, and temperature variance (purple), $\overline{t'^2}$, normalized by a characteristic temperature $T_* = Q_*/(c_p \rho U_*)$; (c) buoyancy flux, $\overline{w'b'}$, normalized by $U_*^3/\overline{h_b}$, $h_b$ refers to the mixed layer depth; (d) momentum flux, $\overline{u'w'}, \overline{v'w'}$, normalized by $U_*^2$; (e) velocity variance, $\overline{u'^2}, \overline{v'^2}, \overline{w'^2}$, normalized by $U_*^2$; and (f) the skewness, $\overline{w'^3}/(\overline{w'^2})^{3/2}$. The turbulence statistics are averaged over the last inertial period ($\sim$17 hours) to reduce the effects of inertial oscillation.





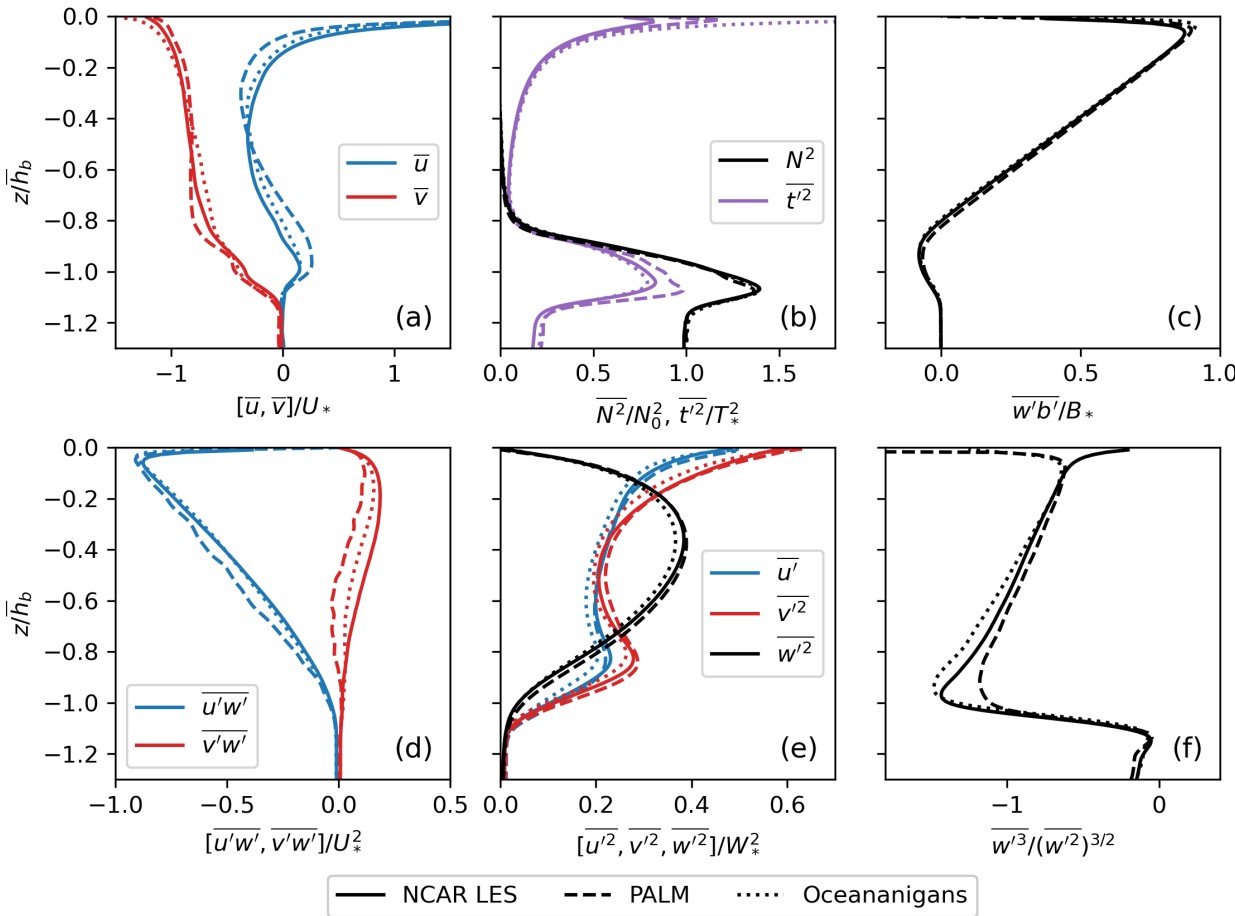

**Figure 15.** Same as Figure 14, except that the turbulence statistics are for a case dominated by convective turbulence, and the buoyancy flux in panel (c) is normalized by the surface buoyancy flux $B_*$ and the three components of the velocity variance in panel (e) are normalized by $W_*^2$ where $W_* = (B_* \overline{h_b})^{1/3}$ is a convective velocity scale.



## 2.6 A rough comparison between vertically-stretched CROCO and NCAR-LES

The preceding detailed comparisons were motivated by a previous set of calculations to compare CROCO to NCAR-LES on 16 previously published scenarios with different combinations of surface wind and cooling in Li and Fox-Kemper (2017). Surface wind and cooling conditions are correspondingly matched for each case. Similar to the accuracy comparisons above, the simulation cases of both CROCO and NCAR-LES models are based on the domain size of $320m \times 320m \times 163.84m$ in $x, y, z$ direction. The computational cells are $256 \times 256 \times 256$ grids in each direction, which corresponds to a horizontal resolution of $dx = dy = 1.25m$. The vertical grids of the NCAR-LES are uniform with a vertical resolution of $dz = 0.64m$. The vertical grids of CROCO, however, were unequal. The CROCO grid points were stretched to be finer near the surface and coarser near the bottom of the domain, as is commonly configured in CROCO and other ROMS applications.

These comparisons spanned a wider range of convective forcing (over a factor of 100) and a wider range of wind stresses (a factor of 4) than the comparisons in the previous sections. The largest differences among the simulations, however, were consistent with the preceding results. According to the comparison of horizontal $(\overline{u'v'})$ and vertical $(\overline{u'w'})$ momentum flux and the turbulence kinetic energy (TKE), the turbulence intensity was slightly weaker in CROCO, which is a similar result to the tendencies in the above comparisons. However, as we have seen in some cases this difference can change according to the SGS scheme and averaging windows. By comparison of buoyancy frequency ($N^2$) and vertical buoyancy flux (w'b'), CROCO had weaker vertical heat transport and entrainment, similar to the differences observed in in Figures 2 and 3 and now these differences can be largely attributed to differences in SGS dissipation rather than numerics. Overall, these comparisons suggest that even in the case of a moderately stretched vertical grid comparable results are to be expected from CROCO as in a uniform grid LES.

## 3 Efficiency comparison

Many factors affect the model computing efficiency, such as the structure and assignment of computing platform, Message Passing Interface (MPI) parallelization, 2D-decomposition of the model and some specific physical parameterizations, particularly ones that have consequences for the stability and allowable time step size. In this section, we compared the computational efficiency of CROCO and NCAR-LES model.

### 3.1 Computing platform – Cheyenne supercomputer

The number and allocation of nodes and processors used for computing and the availability of threads matter to model efficiency. In this study, the Cheyenne supercomputer is used for efficiency tests. The Cheyenne supercomputer, built for NCAR, operates as one of the world's most energy-efficient and high-performance computers. Cheyenne consists of 4,032 dual-socket nodes with 2.3 GHz Intel Xeon E5-2697V4 processors with 18 cores each, for a total of 145,152 cores and a peak performance of 5.34 petaflops. Nodes have either 64 GB or 128 GB of RAM (DDR4-2400) and networked using Mellanox EDR InfiniBand high-speed interconnects with a bandwidth of 25 GBps bidirectional per link. The simulations presented in this paper all ran





on Cheyenne with exclusive use of the nodes. In each efficiency test, the number of nodes, the number of CPU per node, the number of MPI processes and the number of OpenMP threads can be specified.

Combinations of nodes and CPUs per node with different problem sizes and the total number of processors were tested.

When the problem size and total number of processors are fixed, we find that the combination of more nodes and fewer CPUs per node makes the CROCO model compute more efficiently. However, this combination of the selection of nodes and CPUs per node is more costly and so it is typically better to stick to affordable and moderate numbers despite the higher performance, because more nodes requested to Cheyenne make jobs wait longer in the waiting queue, thus the overall time to complete runs is longer though the computing time is shorter.

## 3.2 NCAR-LES 2D-decomposition

NCAR-LES uses pseudo-spectral discretization in the horizontal. Fast Fourier transforms (FFTs) are used to evaluate horizontal derivatives, which requires global data at all grid points in the direction along which the derivatives are evaluated. Thus, a simple domain decomposition in the two horizontal directions would need frequent exchange of large amount of data between different processors, which limits the computational performance. To address this, a 2D domain decomposition is used in

NCAR-LES (Sullivan and Patton, 2008), in which each processor operates on constricted "pencils" that include all the grid points in a specific direction, so that horizontal derivatives along that direction can be evaluated on a single processor with FFT. To evaluate the derivatives on the other direction, a transpose is performed before the evaluation of derivatives, and another transpose is performed afterwards. The combination of transposes and ghost point exchange uses specific communication patterns between only subsets of processors and no global communication is needed. Therefore, large numbers of grid points

can be used and it scales pretty well on thousands of processors. The 2D-decomposition of NCAR-LES is schematized in Figure 16, which illustrates the structure of total number of processors used in the computational process.

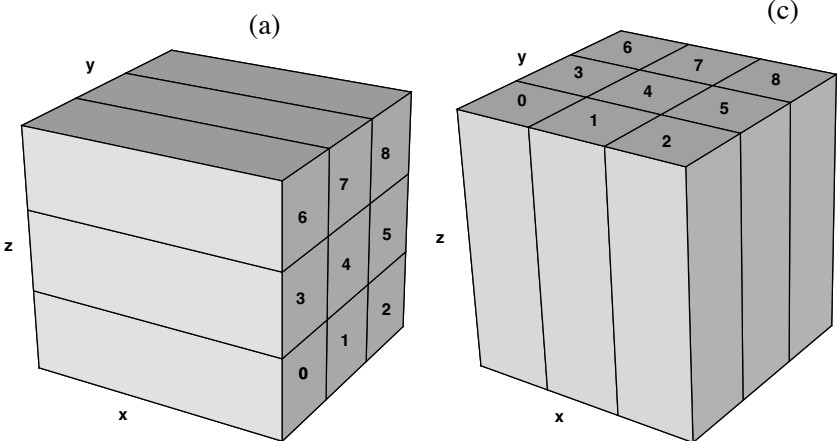

**Figure 16.** 2D domain decomposition on 9 processors: (left) base state with y-z decomposition; and (right) x-y decomposition used in the tridiagonal matrix inversion of the pressure Poission equation (Sullivan and Patton, 2008).





### 3.3 CROCO MPI parallelization

CROCO is currently supported by two parallelization options, MPI and OpenMP, which respectively represent distributed memory and shared memory. The awareness of CROCO MPI or OpenMP settings is necessary to be defined as needed, and the use of MPI or OPENMP is exclusive. According to the test results, when the OpenMP is not called for during compilation in CROCO, the computing time with or without OpenMP threads on Cheyenne does not affect timing so offers no advantages. In this paper, CROCO is used without OpenMP and with MPI, which means only one thread is used for each processor on Cheyenne, and the decomposition of processors and distribution across nodes impact the computing efficiency. The following discussion focuses on the MPI parallelization option.

The structure of CROCO MPI decomposition is divided into XI and ETA direction, NP_XI and NP_ETA in CROCO codes represents the number of processor assignment in XI and ETA horizontal directions respectively. When NP_XI=3 and NP_ETA=3 are set, the MPI parallelization structure is shown in Figure 16c. In order to match the number of processors used in Cheyenne, the product of NP_XI and NP_ETA should be as the same as the product of the number of nodes and the number of CPUs per node should also be matched. Different combinations of NP_XI and NP_ ETA were tested under the frameworks of different combinations of problem size and total number of processors in the Cheyenne environment.

### 3.4 Efficiency test results

The performance of the model efficiency for varying problem sizes and workload per processor is shown from Figure 17 and Figure 18. $NP = NP_z \times NP_x \times NP_y$ where $NP_z$, $NP_x$, and $NP_y$ are the number of processors in the vertical and horizontal directions, respectively. In each figure, the vertical axis is the computing time per time step $t$ multiplied by NP and divided by total work size (i.e., number of grid points or a similar work with a logarithmic multiplier to handle the Fourier pseudospectral costs). $N_z$ is the number of vertical levels, and $N_x$ and $N_y$ are the horizontal grid points.[11]

Figure 17 shows the computational time per grid point for different combinations of problem size (an example of strong scaling). For a given number of total processors NP, the symbol indicates the result found with the most optimal combination of MPI parallelisation or 2D decomposition after experimentation with different parallelism on that problem size. As the number of processors increases, the running time increases under the same problem size reflecting the cost associated with communication among processors. Relatedly, with the same number of processors, a low problem size can take more computing time than a high problem size. The CROCO model shows slightly worse performance per time step on small problems and a potential for better performance in both scaling and cost per time step on large problems. However, given that the timesteps allowed in the NCAR-LES are much larger for these problems than in the CROCO version due to 1) the Boussinesq approximation instead of compressible fluids avoiding limitations of the sound speed, and 2) the numerical choices made in our CROCO setup, the cost per simulated time interval tends to be much higher in CROCO.

---

[11]In Sullivan and Patton (2008), a different scaling for horizontal effort was used because the NCAR-LES is pseudospectral: $M_x = N_x \log N_x$ with $N_x$ the number of grid points in the x direction, or a similar formula for the y direction. This scaling is not used here because CROCO is not pseudospectral.





Figure 18 shows computational time per grid point per slow (baroclinic) time step for a fixed amount of work per processor (an example of weak scaling). The different numbers of barotropic time-steps between each baroclinic time step (NDTFAST) have great influence on computing efficiency in this case. Most intuitively, it can be seen that the runtime of CROCO greatly

increases when NDTFAST is increased, reflecting the cost of additional barotropic time step subcycling. Under weak scaling, the efficiency of the NCAR-LES decreases slightly with larger processor counts. The CROCO efficiency tends toward decreases at first, but then changes in parallelism can recoup some of the losses on high processor counts.

In the weak scaling comparison, different 2D decomposition for models and different nodes and CPU_per_node are used to optimize. The structure of processor distribution are not always the square. So, these aspects may affect the workload/proc and

the comparison results.

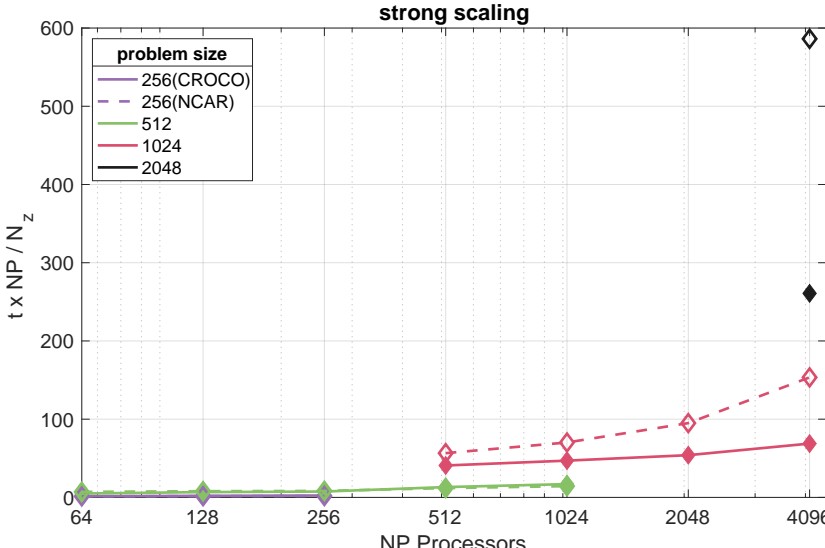

**Figure 17.** Computational time per grid point per time step for different combinations of problem size for CROCO (solid) and NCAR-LES (dashed), an example of strong scaling. NDTFAST=200. a) purple lines and symbols problem size $256^3$; b) green lines and symbols $512^3$; c) red lines and symbols $1024^3$; and d) black symbol $2048^3$..





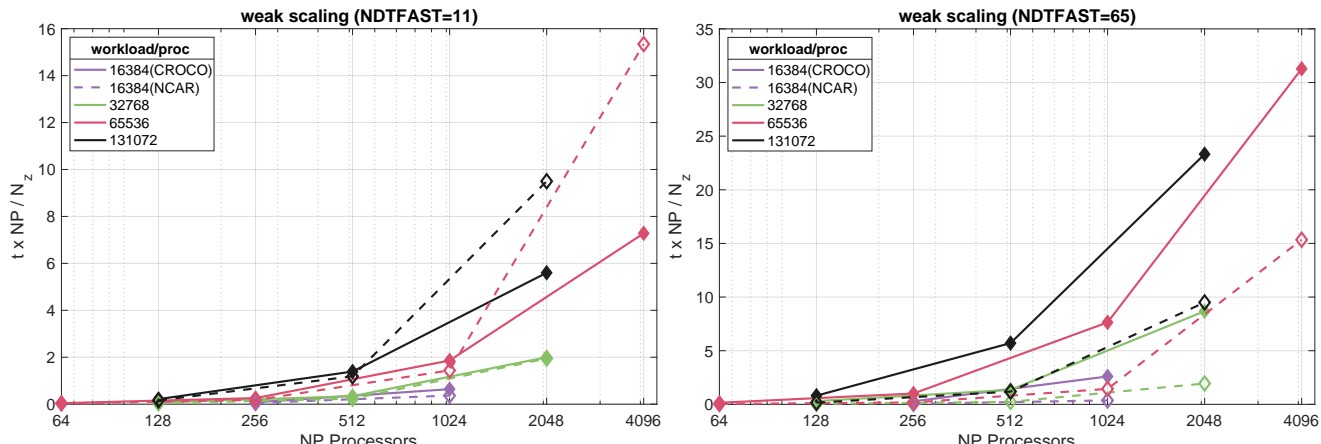

**Figure 18.** Computational time per grid point for a fixed amount of work (i.e., same number of slow time steps and grid points) per processor (an example of weak scaling) with 11 fast (barotropic) time steps per slow (baroclinic) time step (left) and 65 fast time steps per slow time step (right).





## 4    Conclusions

In order to evaluate the performance of the ocean model CROCO with non-hydrostatic kernels, this paper uses NCAR-LES as a benchmark for comparison. The study starts with a comparison of several different LES versions and is recognized for the usability of NCAR-LES. Two comparison aspects of CROCO and NCAR-LES are simulation accuracy and computational
efficiency.

In the accuracy tests, the effect of the explicit SGS parameterization, the second viscosity parameter and the speed-of-sound parameter are varied to understand these key factors impacting simulation accuracy. The NCAR LES results and the CROCO results are overall within expected variations once these effects are considered. For a slightly larger $\overline{u}_0$ in the CROCO runs, the simulated mean flows are very similar. The only notable differences are 1) that the CROCO surface velocity tends to be slightly
higher, 2) that the CROCO surface temperature tends to be slightly lower, and 3) that the CROCO pycnocline entrainment is weaker. These effects are best explained by noting that CROCO's numerical diffusion is weaker than the explicit SGS plus implicit diffusion of the NCAR model. The NCAR runs have stronger internal waves (contributing no buoyancy flux when statistically steady) and less resolved turbulent mixing. There are other minor differences, but most of them are expected due to the different SGS parameterization and limited averaging windows. Overall, the differences between CROCO and the
NCAR-LES are similar to the differences between three different LES codes. The only notable difference that may not be attributable to the SGS parameterization difference is that the NCAR model runs tend to produce more internal waves where higher stratification is present, a difficult to attribute result that is also sensitive to the sound speed setting in CROCO. As for the effect of the second (dilatation) viscosity parameter, increasing $\lambda$ damps marginally unstable modes but allows only moderately larger time steps. Decreasing the speed of sound from $202\,\mathrm{m\,s}^{-1}$ to $3\,\mathrm{m\,s}^{-1}$ allowed a factor of 5 times faster simulations with
negligible changes to the solution accuracy. However, this is a simulation-specific adjustment, and such a large reduction in sound speed is likely to have consequences in other simulated scenarios. A rough comparison between CROCO on a stretched vertical grid and NCAR-LES on a uniform grid, essentially the same explanations of differences apply.

In efficiency tests, based on the Cheyenne supercomputer platform, the difference between CROCO and NCAR-LES on their computational parallelization and 2D-decomposition is elaborated. The relationship of the number of processors and
nodes between model and the computing platforms is tested. As the strong scaling representing the computational time per grid point for different combinations of problem size, and the weak scaling representing computational time per grid point for a fixed amount of work per processor, it is shown that the computational efficiency of CROCO and NCAR-LES per time step is comparable. Increasing the number of barotropic subcycle time steps in CROCO, or increasing the sound speed in CROCO greatly affect its efficiency–easily by a factor of 4 or more.

To sum up, CROCO and LES are comparable on their simulation accuracy and computational efficiency per time step. However, in these idealized test cases where neither compressibility or barotropic flow (where CROCO has specialized capabilities) are important, these capabilities limit the time step in CROCO to be approximately six to fourteen times slower depending on the strength of forcing (or approximately equivalent to a factor of two lower resolution) than the optimal value in NCAR-LES. Optimizations continue: using the Runge-Kutta version of CROCO may allow approximately two times larger timesteps

(Lemarié et al., 2015), avoiding the fifth-order WENO scheme makes CROCO faster (although with larger dispersion and diffusion errors), a new variant on the CROCO nonhydrostatic numerics is in prototype testing and should allow faster simulations, and the version of NCAR-LES used here can be sped up by 10-15% by using a new Fourier transform package (MKL fft). The only differences between CROCO and NCAR-LES that were not attributable to the SGS parameterization differences are that NCAR-LES tends to produce slightly more internal waves. The CROCO solutions are insensitive to the values of the second viscosity and the speed of sound. Therefore, artificially large value of the second viscosity and artificially small value of speed of sound may be used to increase the time-step stably and accurately as long as the speed of sound is faster than the speed of the fastest process that needs to be properly simulated. Overall, when the additional features of CROCO are needed (e.g., nesting, complex topography, etc.), these additional costs can be justified, while in idealized settings the NCAR-LES is faster in arriving at nearly the same result.

*Code availability.* The CROCO ROMS code is available for download at http://croco-ocean.org. The NCAR-LES is available upon request to pps@ucar.edu.

*Data availability.* The simulation data underlying this paper are available on the Brown Digital Repository at https://doi.org/10.26300/vpdb-v266.



*Author contributions.* XF, NS, QL, and BFK conceived the project. XF, NS, and QL ran the simulations. PM, FA, PS contributed expertise

on the CROCO and NCAR-LES models. All authors participated in the writing. XF organized the writing contributions. This project is XF's

master's degree project, and BFK is her advisor.

*Competing interests.* The authors declare no competing interests.

*Acknowledgements.* BFK and XF and computing at Brown University were supported by NSF 1655221. Time on Cheyenne was supplied by

the Computational and Information Systems Laboratory at the National Center for Atmospheric Research (Computational and Information

Systems Laboratory, 2019). QL acknowledges the support from the Center for Ocean Research in Hong Kong and Macau. The collaboration

of NS was supported by Innovation, Information & Biologisation–Fonds ($I^2$B-Fonds).





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
