# Peer review of "Comparison of the Coastal and Regional Ocean Community Model (CROCO) and NCAR-LES in Non-hydrostatic Simulations"

_EGUsphere, 2023_

## Author Comment (AC1)

**Response to Reviewers**

https://egusphere.copernicus.org/preprints/2023/egusphere-2023-1657/egusphere-2023-1657.pdf

We sincerely thank the reviewers for making helpful suggestions.  The changes we will make in response will clearly improve the paper.  We have considered all suggestions, and in most cases will make any suggested changes.  The only exceptions are changes that we could not implement due to some of the simulation data being no longer accessible or changes that would make the paper less amenable to a particular audience, but none of these affect fundamental aspects of the work. You have our sincere appreciation for your invaluable contribution as a reviewer for the paper titled "Comparison of the Coastal and Regional Ocean Community Model (CROCO) and NCAR-LES in Non-hydrostatic Simulations" submitted to *Geoscientific Model Development*.

Your thoughtful and thorough review played a crucial role in enhancing the overall quality of the manuscript. Your constructive feedback and insightful comments significantly contributed to the refinement of the research, ultimately ensuring the paper's academic rigor and relevance. The time and expertise you devoted to evaluating the paper are deeply appreciated. Your commitment to maintaining the high standards of scholarly publishing is instrumental in advancing the field and fostering a culture of excellence.

Red comments below indicate places where the text is to be altered in a revised version of the paper. Other improvements in writing style were made at the same time.  A tracked changes version of the paper will be made available on request.  Discussion and commentary in response to reviewer comments is in black. Reviewer comments themselves are in blue.

**RC1: 'Comment on egusphere-2023-1657', Anonymous Referee #1, 28 Nov 2023**

Review: "Comparison of the Coastal and Regional Ocean Community Model (CROCO) and NCAR-LES in Non-hydrostatic Simulations"

Summary

This is a relatively simple paper that demonstrates 1) many LES codes produce very similar solutions, with most differences attributable to SGS parameterization (or lack thereof), and 2) that CROCO's "pseudocompressible" non-hydrostatic algorithm has significant computational costs (at least apparently, without any further information about differences between the two implementations in software). I think this is good to know and worth publishing. But the presentation has to be improved and especially simplified so that the modest, simple results of the paper are not obscured. These main points just require spending a bit more time on the figures and presentation and don't *require* running any new simulations. That said, I'm also confused why CROCO results are left out of the section 2.5 comparison between PALM and Oceananigans, which adds a convection dominated case that is not included in the NCAR-LES–CROCO comparison, and why the figures (for example 14 and 15) are a little different (but intended to show the same information). This should be cleaned up too.

Thanks for the supportive comments!

This comparison is actually more complex than it might seem, as CROCO was run on a different system than most of the LES runs (with the exception of NCAR-LES), so we didn't do the exact same comparison with LES because the output data was not co-located. Additionally, all of the models in the LES comparison section are Boussinesq approximation, while CROCO is not. Thus we seek a baseline amount of variation among the Boussinesq models before comparing to the compressible CROCO.

Our original intention was to focus on the cases shown in Figures 14 and 15 using only CROCO and NCAR-LES, but we realized that it would be helpful to have a comparison among "accepted" LES schemes first to have a basis for how different CROCO is.  We chose a more idealized set of forcing for that comparison as we wanted to be clear that differences arise from the numerics and SGS of those models, not from the complicated scenario.

Figures 14 and 15 show results for different classes of simulations–i.e., with different surface forcing.  Thus, while they are similar, they do not show the same information as the preceding

comparisons, which are more idealized.  Nonetheless, in response to other comments below we plan to improve the presentation and discussion of all of these figures in revision.

Minor comments

Line 5: "code base" is awkward

Reply:

"code base" is deleted. The sentence will be " Here the Coastal and Regional Ocean COmmunity model (CROCO) and the NCAR Large-Eddy Simulations (NCAR-LES)  models are compared with a focus on their simulation accuracy and computational efficiency. "

Line 9: "To gauge how far CROCO is from NCAR-LES…" is rather vague. This sentence should be improved to more clearly explain why it's useful to bring PALM and Oceananigans into the comparison. In reality, I think this paper could be written without the additional solutions. The additional solutions are useful, however, to construct a useful notion of "accuracy" in the context of typical ocean LES solutions.

Reply: We thank the reviewer for this suggestion.  We now explicitly state that the LES comparison has the purpose of "defining the notion and magnitude of accuracy for the LES vs. CROCO comparison". We also clarify that PALM, Oceananigans, and NCAR-LES are all non-hydrostatic Boussinesq models, while CROCO is a compressible fluid dynamics model.  We have added a sentence to clearly state this intention.

Line 10: "Oceananigans" is misspelled

Yes, "Oceanigans" is changed to "Oceananigans" throughout.

Line 13: The difference in computational costs between CROCO and NCAR-LES should be stated explicitly here. On line 452, it is stated that a CROCO simulation costs between 6-14x more than an NCAR-LES simulation for an idealized case. These hard numbers are some of the most important results of the paper and "order of magnitude" is unnecessarily vague.

Reply: We have clarified the computational costs in these recommended locations and added it to the abstract. Due to subtle differences between the efficiency of simulations under discussion at different points in the paper (e.g., as the sound speed changes), these comparisons are not consistent between CROCO and LES everywhere in the paper. However, we appreciate the reviewer's comment that this is an important result, and we have clarified the key comparisons and elevated the result to the paper abstract.

Line 30: I don't know what ROMS_AGRIF means, exactly, so it may be useful to define this more explicitly and perhaps include a citation.

Reply: The reference in the first sentence using ROMS_AGRIF links to "Debreu et al. (2012)" which is explicit about the ROMS_AGRIF vs. CROCO_ROMS codes. We have clarified that ROMS_AGRIF is a different version of the ROMS modeling system, and details about the differences can be found in Debreu et al. (2012).

Line 30: What is "SNH"?

Reply: We now replace the acronym SNH everywhere with its full meaning, "shallow-water nonhydrostatic".

Lines 47-52: There are a lot of problems with the English and punctuation here. Equation 1 is floating. On line 51, the sentence "Generally, non-hydrostatic ocean modelling is taken on in models that employ the Boussinesq approximation, which result at leading order in incompressible velocities" is hard to decipher.

Reply: We have improved the writing of this passage. It will read:

The addition of a non-hydrostatic solver is a rare feature to incorporate into a coastal model such as CROCO, but some applications on small-scale coastal dynamics will require nonhydrostatic capability. The scalings of the fluid equations for common oceanographic problems (e.g., McWilliams, 1985) indicate that the dimensionless vertical momentum equation has two key parameters determining if hydrostasy will be adequate: the aspect ratio and Froude number (ratio of vertical shear to buoyancy frequency).

$$\underbrace{\frac{H^2}{L^2}}_{\text{aspect}^2} \underbrace{\frac{V^2}{N^2 H^2}}_{\text{Froude}^2} \frac{Dw}{Dt} = -\frac{\partial \phi}{\partial z} + b$$

When non-hydrostatic effects are important, the aspect ratio approaches 1 and the stratification is not stronger than the shear, so the resulting turbulent motions are nearly isotropic.

$$\text{Hydrostatic if: } \frac{H}{L} \ll 1, \qquad \text{Non-hydrostatic if: } \frac{H}{L} \sim 1 \text{ and } \frac{V}{NH} \sim 1$$

Ocean LES are usually used in the non-hydrostatic regime, and thus these models solve the non-hydrostatic equations.

Typically, non-hydrostatic ocean models also employ the Boussinesq approximation (e.g., Marshall et al., 1997). In CROCO, the implementation of non-hydrostatic physics takes advantage of compressible fluid dynamics to arrive at a simplified numerical implementation. In CROCO, the degree of compressibility can be varied by changing the sound speed in the model, but it cannot be chosen to be infinite (i.e., incompressible). Importantly for this paper, the sound speed does not need to be realistic in order to simulate conditions similar to those in non-hydrostatic, Boussinesq approximation LES. The lower the sound speed is, the larger the timesteps can be in CROCO, and thus the more efficient the model becomes. Section 2 explores the sensitivity of CROCO results to changing the sound speed and other parameters that arise only in compressible fluid models.

Figure 1: I don't find this type of 3D surface visualization to be informative: the same information could be conveyed with a heatmap or contour plot. Also, vertical velocity may be a better choice than horizontal velocity because of the presence of mean shear.

Reply: We disagree, after making many of these figures, readers in our group and peers found this figure to be helpful. The lead author actually made an art project based on this figure by 3D computer-controlled milling. We found after sharing the paper with others in our lab group that without this figure the group had a hard time visualizing what the many line plots of statistics represented.

Lines 83-96: What is the buoyancy frequency? There are a lot of parameters listed here that are irrelevant for the physics: thermal expansion coefficient, temperatures, reference density, heat capacity (except for the relationship between the surface heat flux and buoyancy flux)… it might be simpler to simply give the buoyancy frequency and depth of the mixed layer.

Reply: We now provide the buoyancy frequency in what was L90. As the other parameters are important for reproducibility and for understanding figures with units of potential temperature, we leave them as is.

Line 125: Why does CROCO require a slow time-step 14 times slower than NCAR-LES? It's important that the typical implementation of 3rd order Runge-Kutta requires 2 tendency evaluations (and 2 pressure solves) per time-step. Other schemes can require fewer tendency evaluations per time-step. But 3rd-order Runge-Kutta permits higher CFL, to the point that one usually gets faster time-to-solution with 3rd-order Runge-Kutta. From the standpoint of computational efficiency

Reply: It is not only the time stepping that differs in this case, it is the formulation of the equations of motion as compressible in CROCO while they are incompressible/Boussinesq in the other models. This causes an additional equation to be solved for density and short time step limits due to the Courant condition for the speed of sound. We apologize that this was not clear

to the reviewer in the first version, but we have now revised the presentation of that rationale in response to the reviewer's comment about lines 47-52.

Line 130: How does the solution change when the pseudo sound speed is too slow? Are there distinctive qualitative changes to the solution that alert users to a possible issue? This is a key piece of useful information for future users of CROCO that should probably be included in this section.

Reply: This is addressed in Section 2.4. A reference to this discussion is now added to L130.

Section 2.1.1. Why are these scalings / non-dimensionalizations useful? They greatly complicate the comparison while introducing no obvious benefit as far as I can tell; aside from being used as a device to crowd many lines into figures 2 and 3 (more on that later), they are only mentioned in a footnote. The physics discussion here is also basically unrelated to the rest of the paper. It's quite difficult to interpret figures 2 and 3 with all four cases shown anyways (a total of 8 lines that must be picked apart by eye) — it's probably better to have just two lines per plot.

Reply: Since Monin and Obukhov (1954) developed the similarity theory (abbreviated MOST), these nondimensional parameters are commonly used in boundary layer turbulence studies, which we expect may be many of the readers of this paper. We use MOST here to connect to that literature and to recapture if model solutions diverge in both dimensional and dimensionless aspects. Because the turbulent layer depth differs between the models and these non-dimensionalization depends on that depth, $z_w$ found in equation (6), the dimensional and dimensionless comparisons differ when $z_w$ differs among the models.

Line 188: I wish that the vertical axes of figure 2 were not normalized by z_p. Otherwise, it would be far more obvious that the two models produce different deepening rates.

Reply: As in the previous response, we are comparing the dimensionless evolution in Fig. 2, which is theoretically supposed to be controlled by the surface forcing identically (MOST).

Line 188 / footnote 6: The fact that N^2 is the same but the normalization is different, leading to a discrepancy in figure 2d, is very confusing.

Reply: Again, this is meant to illustrate that the dimensionless rate of entrainment differs importantly between the models. According to MOST, this should not occur, so it is an ideal way to illustrate the differing numerical accuracy in a well-studied physical regime. Note that the reason why we selected "classic" boundary layer turbulence for our test cases was to take advantage of the deep understanding of this class of nonhydrostatic physics due to studies using MOST.

Line 192: Okay, but NCAR-LES seems to have stronger vertical velocities (figure 3c) and both models apparently have the same N^2. So how is the resolved w'b' smaller for NCAR-LES? Also, I would expect a model with stronger SGS diffusion to exhibit a smoother velocity field. Or is it only the tracer diffusion that is stronger for NCAR-LES, while the SGS viscosity is weaker than CROCO's implicit viscosity?

Reply: Because NCAR-LES and CROCO differ in the accuracy of their vertical advection schemes for buoyancy (2nd order centered vs. 5th order WENO), it is not as simple as saying that they have the same $N^2$ and $w^2$ and thus must have the same <w'b'>. This is particularly true at the base of the mixed layer, where a sharp change in $N^2$ triggers the WENO monotonic scheme to have substantial effects but these will not occur in the NCAR-LES (which doesn't have this feature). Similarly, a sharp jump is also likely to differ in results between a 2nd and 5th order scheme in any case. The difficulty in capturing entrainment numerically is likely why we see the biggest distinction between models in this aspect.

Furthermore, for reasons we could not diagnose, NCAR-LES has stronger internal waves, which leads to stronger vertical velocities that do not irreversibly transport buoyancy as these are reversible motions instead of irreversible transport. We note this distinction in multiple locations in the paper.

Lines 194: How does CROCO exhibit "limited" third-order dispersion errors, if it uses a 5th order advection scheme? Perhaps this is a reference to the fact that a "5th order" advection scheme is not truly 5th order unless the reconstruction is multi-dimensional, rather than dimension-by-dimension (thus both NCAR-LES and CROCO are formally 2nd order with 3rd order dispersive errors). I'm not sure. Either way, the comment is cryptic.

Reply: This sentence will be rephrased to more accurately characterize the WENO5 errors in the vertical momentum equation.

Note that the NCAR model also has only second-order advection in the vertical with upwinding, so even though it is centered it may have higher-order diffusion and dispersion effects, while CROCO has fifth-order vertical advection with implicit diffusion entering only at the highest orders.

Figures 2, 3, 4: As mentioned above, I think the arcane non-dimensionalizations obfuscate the interpretation of these important figures. Moreover, too much information is displayed. The main points of the paper can be made by comparing CROCO and NCAR-LES solutions for the single case u* = 0.012 m / s and Q* = 50 W / m^2. If showing just two cases, I don't think there is a need to scale z by z_p, or to scale the other variables. Also, I don't think that every covariance needs to be shown. Probably u, v, T, N^2, w'b', and w'w' are enough to make the main points of this section convincingly. If minor points need to be made about the other variables and cases, those figures can be shown separately or in an appendix. More broadly, what information is added by showing the four cases, versus just one?

Reply: We are interested in the response under MOST scaling, as that is the best understood regime of boundary layer turbulence. Secondly, we consider not just wind forcing and

convection forcing together, but separately.  This is because the discretization differs in the momentum and buoyancy equations, and thus their characteristic errors will be different in different combinations.

Figures 5-6: These panels show a lot of redundant information. The two figures should be combined and should illustrate the main point (that NCAR-LES has more energy at high wavenumbers?) with just 2-3 panels.

Reply: We disagree.  NCAR-LES has more along-wind velocity variance, and vertical velocity variance at depth, but in the cross-wind direction there is good agreement until the deepest depths.  This is revealing of the internal gravity wave explanation that we discuss in the text, which relates the different velocities to one another.  At mid-depths, the errors are in the x-z plane while at greatest depths they are more isotropic.  Since the vertical coordinate is stretched in CROCO, this is also important to examine (i.e., where the coordinate stretching is occurring near the surface, there are not big differences). Thus, we prefer to keep these figures.

Line 260: Do we have any idea whether the additional SGS flux in NCAR-LES is realistic or not? Note: Pressel et al. 2017 makes a very similar point in the context of atmospheric LES.

Reply: The NCAR-LES SGS (Sullivan et al. 1994) is a well-trusted scheme that very carefully adheres to the MOST similarity properties by design, and we are simulating in a regime where MOST should apply.  By contrast, CROCO has implicit dissipation at high order and a novel boundary condition scheme that requires compressible fluid dynamics.  Hence the purpose of this paper to compare the two.

Section 2.3: It would be even more useful to know when these numerical parameters _do_ have an effect (and what that effect is). Ditto for section 2.4. If the conclusion of these sections is just "the results are unchanged", I do not think that we need figures 11 and 12, or separate sections 2.3 and 2.4.

Reply: As the second viscosity is only used in compressible fluid dynamics but is being used here in an approximately Boussinesq (weak compressibility) scenario, there is no expectation for when it will become important. As this parameter is only present in CROCO, it was important to explore. However, as the reviewer suggests we can significantly shorten this section. We have now removed the two figures and retained only the first and last paragraph of Section 2.3 and about half of the text and both figures from Section 2.4, but for clarity in describing the experiments carried out we preserve the different sections.

Section 2.5: Why are CROCO results not included in this section? I interpreted this paper as being primarily useful in validating and benchmarking CROCO (not NCAR-LES, which boasts an extensive literature already and is less useful for realistic problems as noted…)

Reply: These simulations were carried out at a different computing center where CROCO was not ported. We think that this section provides important context as to the magnitude of the differences between CROCO and NCAR-LES. However, we did not think it was necessary to include CROCO into these comparisons, as they are only for "defining the notion and magnitude of accuracy for the LES vs. CROCO comparison".

Line 310: Do Oceananigans and PALM also exhibit reduced mixing if explicit SGS diffusion is omitted, as for CROCO? Presumably, this would be a key finding to support some of the main conclusions of section 2.

Reply: We cannot turn off explicit SGS in PALM due to the formulation of that model. It is possible in Oceananigans though. Here is a comparison between AMD SGS scheme and no SGS in Oceananigans with 9th order WENO in a wind driven shear turbulence dominant regime

(). Most of the turbulence statistics look similar. It is unclear if implicit LES results in reduced mixing in Oceananigans — $w^2$ is actually bigger without explicit SGS. But note that these are resolved statistics so the comparison for w'b', w'u' and TKE might not be fair. So I think we don't have enough evidence yet to show the difference between explicit and implicit SGS. It probably depends on the SGS and advection schemes as well as the test case we are looking at.

Section 2.6: This section doesn't seem to add much.

Reply: This section provides a bit of information in simulations more typical of the regime where CROCO would be used in practice (i.e., coastal settings with both stress and convective forcing). That is, the MOST regime is not the pragmatic usage regime for CROCO. We feel that this short section is a helpful reminder to the reader of why CROCO is valuable even though it is more expensive than the other LES codes.

Line 332: What preceding detailed comparisons? Only four cases have been shown.

Reply: We have removed the word "detailed".

Line 411: How much higher is the cost per slow time-step? The results should be stated plainly (they are written later on line 452).

Reply: We have reproduced the cost per slow time step in this location, subject to the configurations in this comparison.

**RC2: 'Comment on egusphere-2023-1657', Anonymous Referee #2, 15 Jan 2024**

Review: "Comparison of the Coastal and Regional Ocean Community Model (CROCO) and NCAR-LES in Non-hydrostatic Simulations"

Thank you for a very interesting manuscript.

Summary

The paper is mainly technical, comparing different LES codes and CROCO from a result and computational performance perspective. It can be seen that the results are quite similar but also that the models are tunable. Are the authors using a standard tuning or have they tuned the models during the study and then present the results for the runs with the closest results and the settings during these runs? The paper would improve if it was easier to see what is a standard setup and what has been tuned in order to get similar results.

The study would also have gained confidence if the results were compared with observations if available or chosen a test case where observational data is available.

The study uses different scaling in the comparison's plots. There are, however, different ways of scaling these runs and the scaling as presented make the analysis rather more difficult than easier to interpret.

Reply:

We use standard tunings for all models, as we explain on Section 2.1.1. In fact, many of the difficulties in arriving at a clean comparison, especially of the relative computational costs, involved retuning parameters to see if they made a significant difference in cost without degrading accuracy.

Unfortunately, as these are idealized forcing cases, there are not meaningful observations to compare against. Instead, we try to restrict our simulations to where the Monin-Obukhov

Similarity Theory (MOST) applies, as it has been found successful in explaining both simulations and observations.

The scalings used to remove the dimensions from the axes in some plots (e.g Figures 2 and 3) are standard approaches for the boundary layer turbulence literature (i.e., MOST). As those figures refer to topics discussed in this literature, we prefer to keep that scaling and to reach that audience of readers. Since the forcing is constant ($u^*$, $w^*$), most of these scaling factors are also just constants except for the boundary layer depth and buoyancy frequency which sometimes vary importantly among the models.

Minor issues

Line 1-15. The abstract would gain of being more precise and using numbers of acceptable, reasonable, or negligible deviation and additional cost of running CROCO compared to NCAR-LES.

Reply: We have added some quantification of cost and accuracy to the abstract. It now reads:

Largely due to the compressible fluid equations it solves, this version of CROCO is found to require six to fourteen times shorter timesteps than NCAR-LES, depending on forcing, and between ½ and 2x the cost per timestep depending on how many barotropic subcycles per baroclinic timestep are used.

Line 5 I assume that "code base" is used instead of model since a model can also include a specific set-up, which is good. However, it is easier to rather change "code base" to just "code".

Reply: We have removed this phrase.

 Rather large differences in Figure 2 d) (N$_2$) and e) ( $\langle$b'w' $\rangle$ ) with deviation larger than 20% which can be compared with the quality assessment set by the authors of some 10% "up to about 10% should be considered negligible". Please discuss these differences.

Reply:

The discussion on Line 188 and near Eq. 7 & Line 155-167 address the fact that entrainment rate is very different between these models, which explains most of the differences in this comparison. The discussion of the different numerical schemes in the vertical advection equation is intended to address the reasons why this entrainment differs.

Figure 1. It is in a way nice to get a visual interpretation of the spatial velocity scales but please add a diagram showing the mean and the variance of the velocity as a function of depth as well.

Reply:  Figures 2 and 3 show the mean and variance of the velocities.

Line 121 spell check "clost". Please discuss why CROCO isn´t stable at CFL time step?

Reply: This has been corrected to "closest".

Line 216 It is noted that evaluation of different numerics in CROCO is possible, but beyond the scope of the paper. It is, however, comparisons with experimental data that really is missing.

Reply:  Unfortunately, these numerical simulations are for idealized forcing settings for which there is no observational or experimental validation.  The use of simulations where MOST applies is the closest we can get to having objective "truth" to compare against.  Mainly, we just have to be content with measuring the difference between the models without knowing which is most accurate.

Line 219-225 Figure 5 and 6 are discussed but the discussion of the discrepancy of the low-wavenumber tails is missing.

Reply: We added the following text to the discussion of these figures: The deviations at low wavenumber are due to the integral constraints of <w>=0 and buoyancy anomaly over the whole domain being linked to vertical fluxes. Thus, the small-scale deviations and large scale deviations are linked. In u' and v', there are not meaningful large-scale deviations.

Figure 14-15 these seem to be plotted using another plotting set-up than previous. Please use the same. Purple and black are very similar in b) plots. Please use a color map with larger differences

Reply: We have redrawn these figures to look more similar to other figures.

Line 329 It´s understandable that there can be larger differences for the cross-wind velocity component v'$_2$ than the along-wind component in the Figure 14e. It´s though surprising that the sign of gradient towards the surface differ. Please elaborate this further.

Reply: We ended up rerunning the Oceananigans calculation to regenerate figure 14. That model is under intensive development, and now the gradient of $v'^2$ seems to be consistent among the three LESs near the surface. We were using an older version of Oceananigans in the previous version of these figures. The new figures are from Oceanaingans v0.83.0. We now include the version of Oceananigans in the paper as it has been under intensive development. We suspect that the boundary condition implementation has been revised during the review of our paper. Note that Oceananigans is not the focus of this paper, CROCO is, so we do not feel the need to go deeper into these questions in the text.

Line 332-333 Difficult to understand why these previous comparisons have motivated this study. Was it motivated since there were so large discrepancies or so small ones. If section 2.6 is to be kept in the paper the rationale needs to be clearer specified.

We now add the following:  In reproducing Li and Fox-Kemper (2017) with both NCAR LES and CROCO, there were notable differences.  However, in that comparison many parameters differed between the models (e.g., stretched vertical grid, subgrid model) in addition to the numerics.  Hence, a more detailed comparison where gridding was more tightly matched and subgrid schemes were explored was carried out (preceding subsections in Section 2).  In this final subsection, a comparison between CROCO and NCAR LES in more typical configurations (where they are not matched in gridding and subgrid schemes) are shown to illustrate discrepancies under more realistic configurations.

Line 366-369 What is meant with "…compute more efficiently"? Is it runtime as a function of processor hours or is it a function of node hours? If it´s per processor hours and not node hours it is questionable to call it more efficient since I assume that the complete node any way is assign this run. Is the above described using "costly" or what is meant here? Is it really the queue that is the problem of is that the total node hours increase? Both efficient above and costly here needs to be defined.

We have rephrased:

When fewer processors per node are used, most systems still typically charge for the unused processors on each node so this is not more efficient overall, just more efficient per processor in use.

Section 4 Conclusions: Here NCAR-LES, NCAR LES and just LES are used for seemingly the same thing. Please correct.

Reply:

We now use "NCAR-LES" consistently throughout the paper in most sections including Section 4, and "LES" for NCAR, Palm or Oceananigans in that section.

Line 422-425 Change order of the second and the third sentence. Something awkward with the second sentence - what is meant here?

Reply:

Fixed. We replace the second sentence with: "The study begins with a comparison of several different LES versions and then because of their close agreement only NCAR-LES is used elsewhere."

Line 428 What's the meaning of "once these effects are considered"?

Reply: We altered this sentence to be clearer:

Once these parameters are considered, the NCAR-LES results and the CROCO results are overall within expected variations.

Line 441-442 Some word is missing in the sentence. Assumen a "In" in the beginning.

Reply: We corrected this sentence to:

A rough comparison between CROCO on a stretched vertical grid and NCAR-LES on a uniform grid finds that the stretched grid does not significantly magnify the model discrepancies in this setting.

Line 449 change "–" to ","? Assume that the factor 4 depends on the amount of change. Could it be expressed differently as a factor of time step or similarly?

Reply: Changed "–easily by a factor of 4 or more" to "by a factor of 4 or more, but it ranged from 2 times to half as expensive as NCAR-LES per time step using a sound speed with accurate results depending on the amount of barotropic subcycling.". It is not easy to express this as a function of fast cycle time step as the model cost is a combination of baroclinic time steps taken and barotropic time steps taken with a very different weighting factor for each. In general, due to

the comparative complexity of the CROCO numerics vs. any of the LES, it is hard to be precise about the costs in a meaningful and fixed way.

Line 454-458 "Optimizations continue:…" Although interesting this comes abrupt in the summary since it has not really been discussed earlier in the paper.

Reply:

Yes, the authors in our group who are working on this effort at present were eager to include this phrase. We now add "which will be documented in future publications" to that sentence to clarify, and rewrote the sentence to lay out these plans and ongoing work more clearly. As this is future work, we think it should only be mentioned in the conclusions section and not elsewhere in the paper as is standard practice.

An Acknowledgments statement thanking the reviewers for improving clarity has been added.